# Learnability of high-dimensional targets by two-parameter models and gradient flow

**Dmitry Yarotsky**
Skoltech
d.yarotsky@skoltech.ru

## Abstract

We explore the theoretical possibility of learning $d$-dimensional targets with $W$-parameter models by gradient flow (GF) when $W < d$. Our main result shows that if the targets are described by a particular $d$-dimensional probability distribution, then there exist models with as few as two parameters that can learn the targets with arbitrarily high success probability. On the other hand, we show that for $W < d$ there is necessarily a large subset of GF-non-learnable targets. In particular, the set of learnable targets is not dense in $\mathbb{R}^d$, and any subset of $\mathbb{R}^d$ homeomorphic to the $W$-dimensional sphere contains non-learnable targets. Finally, we observe that the model in our main theorem on almost guaranteed two-parameter learning is constructed using a hierarchical procedure and as a result is not expressible by a single elementary function. We show that this limitation is essential in the sense that most models written in terms of elementary functions cannot achieve the learnability demonstrated in this theorem.

## 1 Introduction

Starting from the works of Cantor [Cantor, 1878], it is well-known that all finite-dimensional (or even countably-dimensional) real spaces are equinumerable and so, in principle, a set of several real numbers is as descriptive as a single number, or in other words multi-dimensional vectors can be represented by scalars. The idea of reduction of higher-dimensional descriptions to lower-dimensional ones has since appeared in many mathematical works. A couple of notable examples are continuous space-filling curves that fill the whole $\mathbb{R}^d$ [Peano, 1890, Hilbert, 1891] and the Kolmogorov-Arnold Superposition Theorem (KST, Kolmogorov [1957]) that states that any multivariate continuous function can be exactly represented in terms of compositions and sums of finitely many univariate continuous functions.

In the context of machine learning, these results suggest that models with a small number of parameters can potentially be used to represent or approximate high-dimensional objects. In particular, Maiorov and Pinkus [1999] give an example of neural network that has a fixed number of weights but can approximate any continuous function:

**Theorem 1** (Maiorov and Pinkus 1999). *There exists an activation function $\sigma$ which is real analytic, strictly increasing, sigmoidal (i.e., $\lim_{x \to -\infty} \sigma(x) = 0$ and $\lim_{x \to +\infty} \sigma(x) = 1$), and such that any $f \in C([0,1]^n)$ can be uniformly approximated with any accuracy by expressions $\sum_{i=1}^{6n+3} d_i \sigma(\sum_{j=1}^{3n} c_{ij} \sigma(\sum_{k=1}^{n} w_{ijk} x_k + \theta_{ij}) + \gamma_i)$ with some parameters $d_i, c_{ij}, w_{ijk}, \theta_{ij}, \gamma_i$.*

Refinements of this result are given by Guliyev and Ismailov [2016, 2018a,b]. While Theorem 1 contains a non-explicit function $\sigma$, Boshernitzan [1986], Laczkovich and Ruzsa [2000], Yarotsky [2021] give examples of fully explicit fixed-size analytic expressions that also can approximate arbitrary continuous functions. KST has inspired many other results on expressiveness of machine learning models, see e.g. Montanelli and Yang [2020], Schmidt-Hieber [2020], Kůrková [1991, 1992],

38th Conference on Neural Information Processing Systems (NeurIPS 2024).

Köppen [2002], Igelnik and Parikh [2003]. The idea of space-filling curves is used in recent works on generating higher-dimensional distributions from low-dimensional ones [Bailey and Telgarsky, 2018, Perekrestenko et al., 2020, 2021].

While the model appearing in Theorem 1 looks like a standard neural network (apart from the special activation), the proof of its universal approximation property has nothing to do with the method of gradient descent (GD) invariably used nowadays to train neural networks. The proofs of the universal approximation property in this and similar theorems (including the classical universal approximation theorems of Cybenko [1989], Leshno et al. [1993] that consider neural networks with a growing number of neurons) normally consist in presenting, or demonstrating existence of, parameters making the model output arbitrarily close to the target $f$. There is no guarantee whatsoever that these parameters can actually be learned by GD. Moreover, learning by GD is especially problematic for models with a small number of parameters.

In this regard, note that modern deep neural networks are typically abundantly parameterized, with the largest models containing hundreds of billions of weights [Brown et al., 2020, Smith et al., 2022]. One obvious reason for that is the necessity to store a substantial amount of information. But another, more subtle property of large models is that they are easier to train by GD-based optimization [Choromanska et al., 2015], which can be explained by the optimizer having more freedom in finding good descent directions, in particular evading spurious local minima and saddle points.

A convincing and rigorous demonstration that overparameterization may be beneficial for training is provided by the infinite width limits of neural networks in regimes such as NTK [Jacot et al., 2018] and Mean-Field [Mei et al., 2018, Rotskoff and Vanden-Eijnden, 2018, Chizat and Bach, 2018]. While the number of weights in these limits is effectively infinite, the resulting macroscopic loss surface is relatively simple (even convex after reparameterization, in the NTK case); the GD dynamics is analytically tractable and, under mild assumptions, provably trains the model to perfect fit.

In contrast, if the number of parameters is small, then the loss surface tends to be rough and GD inefficient [Baity-Jesi et al., 2018]. Suboptimal local minima are known to be a general feature of finite neural networks with nonlinearities [Auer et al., 1995, Yun et al., 2018, Swirszcz et al., 2016, Zhou and Liang, 2017, Christof and Kowalczyk, 2023]. As the number of parameters is decreased, the chances for GD to get trapped in a bad local minimum increase [Safran and Shamir, 2018].

The above discussion raises a natural abstract question that we address in the present paper:

*Can models with a small number of parameters learn high-dimensional targets by gradient descent?*

In other words, we ask if the possibility of a reduction of a high-dimensional target space to a low-dimensional parametric description reflected in Theorem 1 can at least theoretically be combined with learning by GD, or if this is prevented by fundamental obstacles.

We are not aware of existing rigorous results addressing this question. As remarked, existing results on approximation by highly expressive models do not discuss learning by GD, while publications on GD usually consider standard models such as conventional neural networks. However, we want to address the above question in the most abstract way without assuming any particular model structure. It is clear that models having a small number of parameters and yet GD-learnable, if at all possible, require a very special design.

**Our contribution** in this paper is a resolution of the above question.

1. Our main result is the proof that if the learned targets are represented as $d$-dimensional vectors and are described by a probability distribution in $\mathbb{R}^d$, then there exist models with just $W = 2$ parameters that can learn these targets by Gradient Flow (GF) with success probability arbitrarily close to 1 (Theorem 6).

2. We show that Theorem 6 is actually close to being optimal, since underparameterization with $W < d$ generally implies severe constraints on the set of GF-learnable targets:

   (a) Under a mild nondegeneracy assumption, the GF-learnable targets are not dense in $\mathbb{R}^d$ (Theorem 3). In particular, the success probability cannot generally be made exactly equal to 1 in Theorem 6.

   (b) In contrast, the non-learnable targets are dense in $\mathbb{R}^d$. Moreover, any subset of $\mathbb{R}^d$ homeomorphic to the $W$-sphere contains non-learnable targets (Theorem 4).

(c) The number of parameters in Theorem 6 cannot be decreased to one.

3. In the proof of Theorem 6, the model is constructed using an infinite hierarchical procedure making it not expressible by a single elementary function. We conjecture that the result established in Theorem 6 cannot be achieved with models implementable by elementary functions. For such functions not involving $\sin$ or $\cos$ with unbounded arguments, we prove this as a consequence of the closure of the model image having zero Lebesgue measure in the target space.

We describe the details of our setting in Section 2. In Section 3 we give several general results showing that the underparameterized ($W < d$) learning is theoretically challenging. Then, in Section 4 we present our main result on the almost guaranteed learnability with two parameters. After that, in Section 5 we consider models expressible by elementary functions. Finally, in Section 6 we summarize our findings and discuss several questions that are left open by our research.

Some (more complex or less important) proofs are given in the appendix; in these cases the respective sections are indicated in the theorem statements.

## 2 The setting

In supervised learning one is usually interested in learning *target functions* (or simply *targets*) $f : X \to Y$, with some input and output spaces $X$ and $Y$. We will consider the setting in which the space of targets is a linear space with a euclidean structure. To this end, suppose that $Y$ is a euclidean space with a scalar product $\langle \cdot, \cdot \rangle$, and $X$ is endowed with a measure $\nu$ reflecting the distribution of inputs $\mathbf{x} \in X$ of the function $f$. One can then form the Hilbert space $L^2(X, Y, \nu)$ of functions $f : X \to Y$ equipped with the standard scalar product $\langle f, g \rangle = \int_X \langle f(\mathbf{x}), g(\mathbf{x}) \rangle \nu(d\mathbf{x})$. We will assume that the *target space* $\mathcal{H}$ is a (finite- or infinite-dimensional) subspace of this Hilbert space $L^2(X, Y, \nu)$.

**Examples:**

1. The full space $\mathcal{H} = L^2(X, Y, \nu)$ represents all maps $f : X \to Y$ distinguishable on sets of positive measure $\nu$. If $Y = \mathbb{R}^m$ with the standard scalar product and $\nu = \frac{1}{N} \sum_{n=1}^{N} \delta_{\mathbf{x}_n}$ is an empirical distribution corresponding to a finite subset of $X$, then $\mathcal{H}$ is finite-dimensional, $\mathcal{H} \cong \mathbb{R}^d$ with $d = Nm$. On the other hand, if $\nu$ is not finitely supported, then $\dim \mathcal{H} = \infty$.

2. Let $X = \mathbb{R}^n, Y = \mathbb{R}$, and the targets $f : X \to Y$ be linear functions. If $\nu$ is nondegenerate in the sense that the covariance matrix $[\int x_i x_j \nu(d\mathbf{x})]_{i,j=1}^n$ is full-rank, then the respective target space $\mathcal{H}$ is $n$-dimensional (otherwise, $\dim \mathcal{H} < n$).

3. Let $X = \mathbb{R}^n, Y = \mathbb{R}$, and the targets be polynomials of degree not larger than $q$. Then $\dim \mathcal{H} \leq \binom{n+q}{q}$, with equality attained for suitably nondegenerate measures $\nu$.

We will often write the function $f$ considered as an element of $\mathcal{H}$ as $\mathbf{f}$. Throughout the paper, we refer to the dimension of the target space $\mathcal{H}$ as *target dimension* and denote it by $d$. Note that the target dimension $d$ is not to be confused with the dimensions of $X$ and $Y$ (in fact, $X$ does not even have to be a linear space and have a dimension). Rather, the target dimension $d$ is the number of scalar parameters required to specify a particular target $\mathbf{f}$ within the space $\mathcal{H}$ of considered targets.

Suppose that we are learning the target $f \in \mathcal{H}$ using a parametric model with $W$ parameters. We will view this model as a map $\Phi : \mathbb{R}^W \to \mathcal{H}$ between the parameter space $\mathbb{R}^W$ and the target space $\mathcal{H}$.

We will consider Gradient Flow (GF), i.e. the continuous version of gradient descent. Learning by GF prescribes that the parameter vector $\mathbf{w}$ be evolved by

$$\frac{d\mathbf{w}(t)}{dt} = -\nabla_{\mathbf{w}} L_{\mathbf{f}}(\mathbf{w}(t)), \tag{1}$$

where we use the standard square loss, $L_{\mathbf{f}}(\mathbf{w}) = \frac{1}{2} \mathbb{E}_{\mathbf{x} \sim \nu} \| f(\mathbf{x}) - \Phi(\mathbf{w})(\mathbf{x}) \|_Y^2$, which can equivalently be written as

$$L_{\mathbf{f}}(\mathbf{w}) = \frac{1}{2} \| \mathbf{f} - \Phi(\mathbf{w}) \|_{\mathcal{H}}^2. \tag{2}$$

Here, the subscripts $Y, \mathcal{H}$ on the norms indicate the respective spaces. We will assume for definiteness that GF starts from $\mathbf{w}(t = 0) = \mathbf{0}$.

We will always assume that $W$ is finite and $\Phi$ is differentiable with a Lipschitz-continuous differential. In this case Eq. (1) is, by Picard-Lindelöf theorem, uniquely solvable locally (i.e., for a bounded interval of times). It is possible to relax the Lipschitz differentiability assumption using the special structure of the gradient flow equation (see e.g. the expository paper Santambrogio [2017]), but we will not need this in this work.

In fact, GF (1) is solvable not only locally, but also globally, i.e. the solution $\mathbf{w}(t)$ exists for any $t > 0$. Indeed, the only obstacle for the global existence is the divergence of the solution in finite time, but it is ruled out by the inequality

$$\|\mathbf{w}(t)\|^2 \leq \Big( \int_0^t \|\tfrac{d\mathbf{w}(\tau)}{d\tau}\| d\tau \Big)^2 \leq t \int_0^t \|\tfrac{d\mathbf{w}(\tau)}{d\tau}\|^2 d\tau \tag{3}$$

$$= -t \int_0^t \tfrac{dL_{\mathbf{f}}(\mathbf{w}(\tau))}{d\tau} d\tau = (L_{\mathbf{f}}(\mathbf{0}) - L_{\mathbf{f}}(\mathbf{w}(t)))t \leq L_{\mathbf{f}}(\mathbf{0})t.$$

Let $F_\Phi$ denote the set of targets $\mathbf{f}$ for which the respective GF converges to $\mathbf{f}$:

$$F_\Phi = \{\mathbf{f} \in \mathcal{H} : \inf_t L_{\mathbf{f}}(\mathbf{w}(t)) = 0 \text{ for } \mathbf{w}(t) \text{ given by (1)}\}. \tag{4}$$

Our goal in the remainder of this work will be to examine if we can ensure, by a suitable design of the map $\Phi : \mathbb{R}^W \to \mathbb{R}^d$ with $W < d$, that the set $F_\Phi$ is sufficiently large.[1] We will refer to targets $\mathbf{f} \in F_\Phi$ as *GF-learnable* or simply *learnable*. We remark that, assuming a standard norm topology in $\mathcal{H}$, the set $F_\Phi$ is Borel-measurable as the countable intersection of the open sets $\{\mathbf{f} \in \mathcal{H} : \inf_t L_{\mathbf{f}}(\mathbf{w}(t)) < 1/n, \text{ where } \mathbf{w}(t) \text{ is given by (1)}\}, n = 1, 2, \ldots$

**Example:** To clarify our setting, suppose that we fit to data a linear function $f(\mathbf{x}) = \mathbf{k}^T\mathbf{x}$ with $\mathbf{k}, \mathbf{x} \in \mathbb{R}^d$ for some $d$ (see example 2 earlier in the section). Normally, this function is learned by applying GF to the $d$-dimensional parameter vector $\mathbf{k}$. In our setting, however, we rather rewrite this model in the form $y = \Phi(\mathbf{w})^T\mathbf{x}$, with some generally nonlinear $\Phi : \mathbb{R}^W \to \mathbb{R}^d$, and ask if we can learn the model by using GF w.r.t. $\mathbf{w}$ with $W < d$. In this sense, we decouple the linear weight-dependence from the linear input-dependence and replace it by a nonlinear one.

Note that formulation (1)-(2) of GF is stated purely in terms of vectors and maps in Hilbert spaces, without any reference to the underlying sets $X, Y$ and the measure $\nu$. It is this abstract Hilbert space formulation that we will deal with in the remainder of the paper.

Some of our results, including the main "positive" Theorem 6, require the target Hilbert space to be finite-dimensional, i.e. $\mathcal{H} \cong \mathbb{R}^d$ with $d < \infty$. Others (namely the "negative" Theorems 2, 3, 4) remain valid for infinite-dimensional Hilbert spaces $\mathcal{H}$. By a slight abuse of notation, in this latter case we also write $\mathcal{H} \cong \mathbb{R}^d$, but with $d = \infty$.

## 3 General impossibility results

We start with several general results showing fundamental limitations of GF with a small dimension $W$. First, it is easy to see that one-parameter models can ensure GF convergence only for a very small set of targets.

**Proposition 2.** *Let $W = 1$. Then, if a target $\mathbf{f} \in F_\Phi$, then either $\mathbf{f} = \Phi(w)$ for some $w$, or $\mathbf{f} \in \{\mathbf{f}_-, \mathbf{f}_+\} = \{\lim_{w \to \pm\infty} \Phi(w)\}$ (if any of these two limits exist). In particular, if $\mathcal{H} = \mathbb{R}^d$ with $2 \leq d < \infty$, then $F_\Phi$ has Lebesgue measure 0 in $\mathcal{H}$.*

*Proof.* The first statement follows since the optimization trajectory $w(t)$ is a scalar monotone function of $t$. The second statement on Lebesgue measure follows by Sard's theorem. $\qquad\square$

The next result shows that if $W < d$ and $\Phi$ is sufficiently regular and non-degenerate at $\mathbf{w} = \mathbf{0}$, then $F_\Phi$ cannot be dense in $\mathcal{H}$.

---

[1]Note that if $W \geq d$, then we can easily ensure $F_\Phi = \mathcal{H}$ by taking any surjective linear operator $\Phi : \mathbb{R}^W \to \mathbb{R}^d$ as the model. Indeed, in this case for any target $\mathbf{f} \in \mathbb{R}^d$ the loss function $L_{\mathbf{f}}(\mathbf{w}) = \frac{1}{2}\|\mathbf{f} - \Phi\mathbf{w}\|^2$ is quadratic with the global minimum $L_{\mathbf{f}}(\mathbf{w}_*) = 0$, the GF is solved as $\mathbf{w}(t) = \Phi^T(\Phi\Phi^T)^{-1}(1 - e^{-\Phi\Phi^T t})\mathbf{f}$, and $\Phi\mathbf{w}(t) \xrightarrow{t \to \infty} \mathbf{f}$.

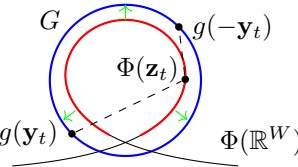

Figure 1: Proof of Theorem 4. GF cannot converge for all points of $G$: such a convergence would require $\Phi(\mathbf{z}_t)$ to be simultaneously close to both $g(\mathbf{y}_t)$ and $g(-\mathbf{y}_t)$, which are far from each other.

**Theorem 3.** *Let $1 \leq W < d \leq \infty$ and $\Phi : \mathbb{R}^W \to \mathcal{H}$ be a $C^2$ map such that at $\mathbf{w} = \mathbf{0}$ the Jacobi matrix $J_0 = \frac{\partial \Phi}{\partial \mathbf{w}}(\mathbf{0})$ has full rank $W$. Then $F_\Phi$ is not dense in $\mathcal{H}$.*

*Proof.* We show that there is a ball of targets for which the GF trajectory gets trapped at a local minimum due to a loss barrier. Without loss of generality, assume that $\Phi(\mathbf{0}) = \mathbf{0}$. Let $\mathbf{f}_0 \in \mathbb{R}^d$ be some length-$l$ vector orthogonal to the range of the differential $J_0$; such an $\mathbf{f}_0$ exists because $d > W$. Let $B_{\mathbf{f}_0,\epsilon} = \{\mathbf{f} \in \mathbb{R}^d : \|\mathbf{f} - \mathbf{f}_0\| < \epsilon\}$. Since $\Phi$ is $C^2$, we have $\Phi(\mathbf{w}) = J_0\mathbf{w} + R(\mathbf{w})$, with a remainder $\|R(\mathbf{w})\| \leq C\|\mathbf{w}\|^2$ for all sufficiently small $\|\mathbf{w}\|$ with some constant $C$. Then, for $\mathbf{f} \in B_{\mathbf{f}_0,\epsilon}$ we have

$$L_{\mathbf{f}}(\mathbf{w}) - L_{\mathbf{f}}(\mathbf{0}) = \tfrac{1}{2}\|\mathbf{f} - \Phi(\mathbf{w})\|^2 - \tfrac{1}{2}\|\mathbf{f}\|^2 = \tfrac{1}{2}\|\Phi(\mathbf{w})\|^2 - \langle \mathbf{f}, \Phi(\mathbf{w})\rangle$$
$$\geq \tfrac{1}{2}\|J_0\mathbf{w}\|^2 - (Cl\|\mathbf{w}\|^2 + \|J_0\|\epsilon\|\mathbf{w}\|) + O(\epsilon^2 + \|\mathbf{w}\|^3).$$

Since $\operatorname{rank} J_0 = W$, the matrix $J_0^* J_0$ is strictly positive definite. Choose $l$ small enough so that $J_0^* J - Cl$ is still strictly positive definite. Then, if we subsequently choose $r$ and then $\epsilon$ small enough, we have $L_{\mathbf{f}}(\mathbf{w}) - L_{\mathbf{f}}(\mathbf{0}) > 0$ for all $\mathbf{w}$ such that $\|\mathbf{w}\| = r$, i.e. for $\mathbf{f} \in B_{\mathbf{f}_0,\epsilon}$ the GF trajectory $\mathbf{w}(t)$ never leaves the ball $U_r = \{\mathbf{w} \in \mathbb{R}^W : \|\mathbf{w}\| \leq r\}$. Then, since $\Phi$ is continuous, if $\epsilon < l$ and $r$ is small enough, $B_{\mathbf{f}_0,\epsilon} \cap \Phi(U_r) = \varnothing$ and so the ball $B_{\mathbf{f}_0,\epsilon}$ cannot be reached by GF. $\square$

Finally, we show that for $W < d$ any subset of $\mathbb{R}^d$ homeomorphic to the $W$-sphere contains non-learnable targets:

**Theorem 4.** *Let $1 \leq W, d \leq \infty$. Suppose that a set $G \subset \mathbb{R}^d$ is the image of the $W$-dimensional sphere $\mathbb{S}^W = \{\mathbf{y} \in \mathbb{R}^{W+1} : \|\mathbf{y}\| = 1\}$ under a continuous and injective map $g : \mathbb{S}^W \to \mathbb{R}^d$. Then $G \not\subset F_\Phi$.*

*Proof (see Figure 1).* We use the Borsuk-Ulam antipodality theorem saying that for any continuous map $\phi : \mathbb{S}^W \to \mathbb{R}^W$ there exists a pair of antipodal points $\mathbf{y}, -\mathbf{y} \in \mathbb{S}^W$ such that $\phi(\mathbf{y}) = \phi(-\mathbf{y})$.

Let $\mathbf{w}_{\mathbf{f}}(t)$ denote the solution of GF (1) with target $\mathbf{f}$. For any $t > 0$, consider the map $\phi_t : \mathbb{S}^W \to \mathbb{R}^W$ given by $\phi_t(\mathbf{y}) = \mathbf{w}_{g(\mathbf{y})}(t)$. By the assumption on $g$ and the continuous dependence of GF on the target, the map $\phi_t$ is continuous. By Borsuk-Ulam, it follows that there is $\mathbf{y}_t \in \mathbb{S}^W$ such that $\phi_t(\mathbf{y}_t) = \phi_t(-\mathbf{y}_t)$. Denote this common output vector by $\mathbf{z}_t$.

Let $l = \inf_{\mathbf{y} \in \mathbb{S}^W} \|g(\mathbf{y}) - g(-\mathbf{y})\|$. Observe that $l > 0$, by the continuity and injectivity of $g$ as well as compactness of $\mathbb{S}^W$. Then for any $t$

$$\sup_{\mathbf{y} \in \mathbb{S}^W} L_{g(\mathbf{y})}(\mathbf{w}_{g(\mathbf{y})}(t)) = \sup_{\mathbf{y} \in \mathbb{S}^W} \tfrac{1}{2}\|g(\mathbf{y}) - \Phi(\mathbf{w}_{g(\mathbf{y})}(t))\|^2$$
$$\geq \tfrac{1}{2}\max\left(\|g(\mathbf{y}_t) - \Phi(\mathbf{z}_t))\|^2, \|g(-\mathbf{y}_t) - \Phi(\mathbf{z}_t)\|^2\right) \geq \tfrac{1}{2}(\tfrac{l}{2})^2 > 0. \tag{5}$$

Now suppose that $G \subset F_\Phi$. Then for any $\mathbf{y} \in \mathbb{S}^W$ the function $t \mapsto L_{g(\mathbf{y})}(\mathbf{w}_{g(\mathbf{y})}(t))$ monotonically converges to 0 as $t \to \infty$. However, since $\mathbb{S}^W$ is compact and $L_{g(\mathbf{y})}(\mathbf{w}_{g(\mathbf{y})}(t))$ is continuous in $\mathbf{y}$, such a convergence must be uniform over $\mathbf{y} \in \mathbb{S}^W$, contradicting the lower bound (5). $\square$

Note that we did not assume that $W < d$ in this theorem, but it is vacuous for $W \geq d$ because (again by Borsuk-Ulam) there are no continuous injective maps $g : \mathbb{S}^W \to \mathbb{R}^d$. On the other hand, there are plenty of such maps for $W < d$, implying in particular the following corollary.

**Corollary 5.** *If $W < d$, then $\mathcal{H} \setminus F_\Phi$ is dense in $\mathcal{H}$.*

Our use of the Borsuk-Ulam theorem is inspired by DeVore et al. [1989] who used it to establish lower bounds on nonlinear $n$-widths in an abstract approximation setting.

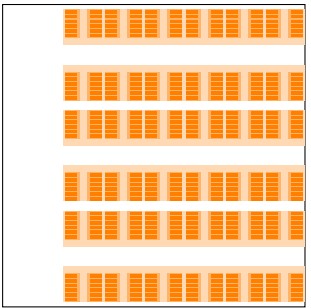

Figure 2: In Theorem 6, we ensure that the learnable set of targets $F_\Phi$ contains a multidimensional "fat" Cantor set $F_0$ having almost full measure $\mu$. The set $F_0$ has the form $F_0 = \cap_{n=1}^\infty \cup_\alpha B_\alpha^{(n)}$, where $\{B_\alpha^{(n)}\}_{n,\alpha}$ is a nested hierarchy of rectangular boxes in $\mathbb{R}^d$. Here, $n$ is the level of the hierarchy and $\alpha$ is the index of the box within the level.

## 4 Almost guaranteed learning with two parameters

Results of the previous section show that for models with $W < d$ parameters there is always a significant amount of non-learnable targets, and models with just $W = 1$ parameter cannot learn sets of targets of positive Lebesgue measure. We give now our main result showing that already with $W = 2$ parameters, one can design maps $\Phi : \mathbb{R}^W \to \mathbb{R}^d$ for which the learnable set is arbitrarily large with respect to a given probability distribution on the target space:

**Theorem 6** (A). *Let $\mathcal{H} = \mathbb{R}^d$ with $d < \infty$, and let $\mu$ be any Borel probability measure on $\mathcal{H}$. Then for any $\epsilon > 0$ there exists a $C^\infty$ map $\Phi : \mathbb{R}^2 \to \mathcal{H}$ such that $\mu(\mathcal{H} \setminus F_\Phi) < \epsilon$.*

**Example:** Suppose again that we learn a linear function $f_\mathbf{k}(\mathbf{x}) = \mathbf{k}^T \mathbf{x}$ with $\mathbf{k}, \mathbf{x} \in \mathbb{R}^d$ for some $d$. Assuming a non-degenerate distribution of inputs $\nu$, the target space $\mathcal{H} \cong \mathbb{R}^d$. Suppose now that possible coefficient vectors $\mathbf{k}$ have, say, the prior distribution $\mu \sim \mathcal{N}(0, \mathbf{1}_{d \times d})$ in $\mathbb{R}^d$. Then Theorem 6 states that for any $\epsilon > 0$ there exists a two-parameter model $\Phi : \mathbb{R}^2 \to \mathbb{R}^d$ such that for all target vectors $\mathbf{k}$ except for a set of $\mu$-measure $\epsilon$, for the GF trajectory $\mathbf{w}(t) \in \mathbb{R}^2$ defined by (1) we have $\lim_{t \to \infty} \Phi(\mathbf{w}(t)) = f_\mathbf{k}$.

We give now a sketch of proof of Theorem 6, illustrated by Figures 2 and 3a-3c.

The key challenge in proving this theorem is to ensure that for a majority of targets $\mathbf{f}$ the GF trajectory will not be trapped at a local minimum. This is difficult because, due to the low parameter dimension, a typical point $\mathbf{w}$ in the parameter space belongs to a large number of optimization trajectories with different targets $\mathbf{f}$, and all these trajectories are controlled by a single map $\Phi : \mathbb{R}^2 \to \mathbb{R}^d$.

The key idea of our construction is to implement an aligned hierarchical decompositions of both the parameter and target spaces so that each element of the hierarchy of target subsets can be served by a respective element of the hierarchy of parameter subsets.

The targets for which we guarantee learnability form a $d$-dimensional Cantor set $F_0$ (product of one-dimensional sets) of almost full measure $\mu$ (see Figure 2). This Cantor set is constructed by a sequence of "carving" (or "splitting") stages. Accordingly, the map $\Phi$ is sub-divided into a sequence of maps $\Phi^{(n)}$ associated with stripes of the $\mathbf{w}$-plane and aligned with the respective carving stages (see Figures 3a-3b).

One of the two parameters, $u$, always increases during GF for targets from $F_0$, and the map $\Phi$ can be described in terms of the "level lines" $l_u = \{(u, v) : v \in \mathbb{R}\}$ in the parameter space and the respective "level curves" $\Phi(l_u)$ in the target space. To define stage-$n$ sub-maps $\Phi^{(n)}$ corresponding to levels $n$ of the target Cantor hierarchy, we choose a discretized sequence $0 = u_0 < u_1 < u_2 < \dots$. The domains of the maps $\Phi^{(n)}$ are then separated by the respective level lines $l^{(n)} \equiv l_{u_n}$. The stage-$n$ map $\Phi^{(n)}$ can be thought of as describing the transformation of the level curve $\Phi(l^{(n)})$ to $\Phi(l^{(n+1)})$.

Each stage $n$ is associated with a particular splitting index $k_n \in \{1, \dots, d\}$. The level curve $\Phi(l^{(n)})$ includes multiple linear segments $\Phi(l_\alpha^{(n)})$ oriented along the $k_n$'th axis and approximately coinciding ("aligned") with certain one-dimensional edges of the boxes $B_\alpha^{(n)}$ that represent the $n$'th level of the Cantor set $F_0$. During each "carving" stage $n$, each box $B_\alpha^{(n)}$ is split into sub-boxes $B_\beta^{(n+1)}$ along the axis $k_n$. At the same time, the map $\Phi^{(n)}$ describes the transformation of the aligned segments $\Phi(l_\alpha^{(n)})$ to the next-level segments $\Phi(l_\beta^{(n+1)})$, aligned with the boxes $B_\beta^{(n+1)}$ along the axis $k_{n+1}$.

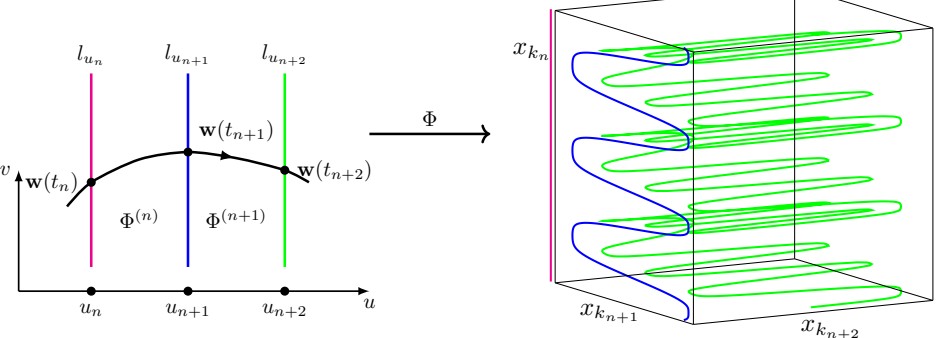

(a) **Stage-wise decomposition of the map** $\Phi$. The map $\Phi$ is defined by its stages $\Phi^{(n)} = \Phi|_{u_n \leq u \leq u_{n+1}}$ separated by the level lines $l^{(n)} \equiv l_{u_n} = \{(u,v) : u = u_n\}$ and respective level curves $\Phi(l^{(n)})$. Each stage $\Phi^{(n)}$ deforms the level curve $\Phi(l^{(n)})$ in the splitting direction $x_{k_{n+1}}$ to form the new level curve $\Phi(l^{(n+1)})$. A non-exceptional GF trajectory $\mathbf{w}(t) = (u(t), v(t))$ passes through all level lines. The splitting indices $k_n$ cycle over the values $1, \ldots, d$ to ensure convergence w.r.t. each coordinate.

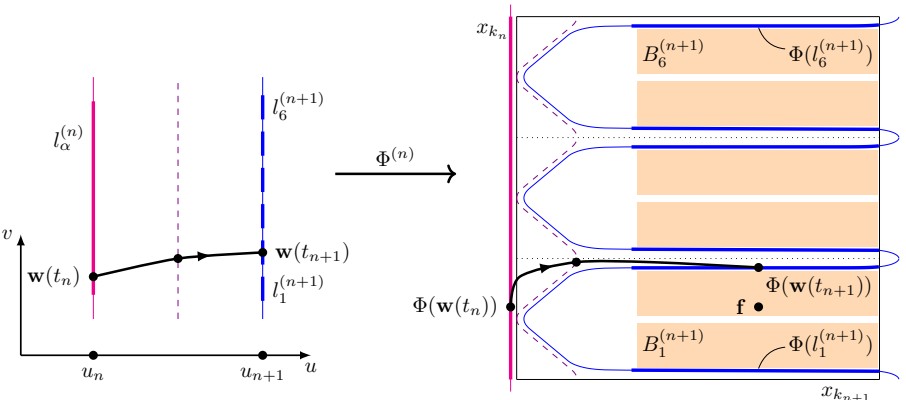

(b) **Box splitting and curve-box alignment at the stage** $\Phi^{(n)}$. The stage curve $\Phi(l^{(n)})$ includes segments $l_\alpha^{(n)}$ (thick red and blue segments) aligned with respective boxes $B_\alpha^{(n)}$ of the box hierarchy. On the right, a box $B_\alpha^{(n)}$ (the big square) is split into $2s_n = 6$ smaller boxes $B_\beta^{(n+1)}$ along the splitting direction $x_{k_n}$. Accordingly, the aligned segment $l_\alpha^{(n)}$ is transformed into 6 new aligned segments $l_\alpha^{(n+1)}$ (thick blue). The splitting only affects the coordinates $x_{k_n}$ and $x_{k_{n+1}}$. During the splitting, gaps are left in the direction $x_{k_n}$ between the child boxes, and in the direction $x_{k_n}$ between the level curve $\Phi(l_\alpha^{(n)})$ and the child boxes, to accommodate convergent GF trajectories. Each non-exceptional GF trajectory $\mathbf{w}(t)$ passes through some aligned segments $l_\alpha^{(n)}, l_\beta^{(n+1)}$.

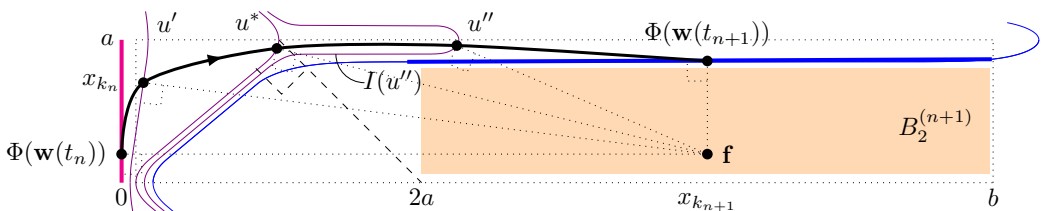

(c) **Transition from** $\Phi(l^{(n)})$ **to** $\Phi(l^{(n+1)})$ through intermediate level curves $\Phi(l_{u'}), \Phi(l_{u^*}), \Phi(l_{u''})$ (violet). These curves ensure that during the $n$'th stage, for all targets $\mathbf{f} = (f_1, \ldots, f_d)$ in the respective box $B_\beta^{(n+1)}$, a point $\Phi(\mathbf{w}(t_n))$ having a coordinate $\Phi_{k_n}(\mathbf{w}(t_n)) \approx f_{k_n}$ is moved by GF to a point $\Phi(\mathbf{w}(t_{n+1}))$ with a coordinate $\Phi_{k_{n+1}}(\mathbf{w}(t_{n+1})) \approx f_{k_{n+1}}$. The points $\Phi(\mathbf{w}(t))$ are approximately those closest to $\mathbf{f}$ on the respective level curves. To avoid local minima, the level curves $\Phi(l_u)$ at each $u$ must be deformed at each $u$ so as to bring such points closer to $\mathbf{f}$. The desired propagation from $\Phi(\mathbf{w}(t_n))$ to $\Phi(\mathbf{w}(t_{n+1}))$ can be achieved by first deforming $\Phi(l_u)$ so as to bring $\Phi(\mathbf{w}(t))$ to the tip of the line $\Phi(l_{u^*})$ ("gathering sub-stage"), and then extending this tip so as to let $\Phi(\mathbf{w}(t))$ slip off it at the appropriate position $x_{k_{n+1}}$ ("spreading sub-stage").

Figure 3: The map $\Phi$ from Theorem 6 (see Section A for details).

During each stage $n$, a part of the box $B_\alpha^{(n)}$ is removed and the map $\Phi^{(n)}$ is adjusted so as to ensure that for each target $\mathbf{f}$ from the resulting Cantor set $F_0$ the GF trajectory goes through some aligned pieces $\Phi(l_\beta^{(n+1)})$ to the very point $\mathbf{f}$. In a particular stage $n$, the GF trajectory "goes around the corner" of the next-level box (see Fig. 3c), so that the agreement of the current approximation $\Phi(\mathbf{w}(t_n))$ with $\mathbf{f}$ in coordinate $k_{n+1}$ is substantially improved at the cost of a slightly degrading agreement in coordinate $k_n$. As the boxes become smaller, the overall disagreement between $\Phi(\mathbf{w}(t_n))$ and $\mathbf{f}$ gradually vanishes.

For general targets in the current box $B_\alpha^{(n)}$, the GF trajectory can get trapped at a local minimum – in particular, if the coordinate $f_{k_{n+1}}$ of the target is close to the respective coordinate of the aligned level piece $\Phi(l_\alpha^{(n)})$. For this reason, some parts of the box $B^{(n)}$ are removed during splitting for the next stage. The total measure of the removed parts can be made arbitrarily small by adjusting splitting parameters. In this way we ensure that all the vectors $\mathbf{f} \in F_0$ are learnable and the measure $\mu(F_0)$ is arbitrarily close to the full measure.

# 5    Models expressible by elementary functions

The model $\Phi$ constructed in Theorem 6 involves an infinite hierarchy of maps $\Phi^{(n)}$ and as a result (and in contrast to conventional models such as neural networks) is not expressible by a single elementary function. It is natural to ask if this non-elementariness is essential or only a feature of our proof. We conjecture it to actually be a necessary feature of models $\Phi : \mathbb{R}^W \to \mathbb{R}^d$ when $W < d$ and the set $F_\Phi$ of GF-learnable targets is sufficiently large, say has a positive Lebesgue measure in $\mathbb{R}^d$.

One setting in which we can prove this conjecture is when the closure $\overline{\Phi(\mathbb{R}^W)}$ of the image $\Phi(\mathbb{R}^W)$ has Lebesgue measure 0. Obviously, this is a sufficient condition for the set $F_\Phi$ of GF-learnable targets to have Lebesgue measure 0. We show that $\overline{\Phi(\mathbb{R}^W)}$ has measure 0 for so-called *Pfaffian* functions known to have strong finiteness properties [Khovanskii, 1991]. Pfaffian functions include all elementary functions, but not necessarily on the largest domain of their definition; most importantly, $\sin$ is Pfaffian only when considered on a bounded interval.

Precisely, a *Pfaffian chain* is a sequence $g_1, \ldots, g_l$ of real analytic functions defined on a common connected domain $U \subset \mathbb{R}^W$ and such that the equations

$$\tfrac{\partial g_i}{\partial w_j}(\mathbf{w}) = P_{ij}(\mathbf{w}, g_1(\mathbf{w}), \ldots, g_i(\mathbf{w})), \quad \substack{1 \le i \le l \\ 1 \le j \le W}$$

hold in $U$ for some polynomials $P_{ij}$. A *Pfaffian function* in the chain $(g_1, \ldots, g_l)$ is a function on $U$ that can be expressed as a polynomial $P$ in the variables $(\mathbf{w}, g_1(\mathbf{w}), \ldots, g_l(\mathbf{w}))$. *Complexity* of the Pfaffian function is determined by the length $l$ of the chain, the maximum degree $\alpha$ of the polynomials $P_{ij}$, and the degree $\beta$ of the polynomial $P$, and so can be defined as the triplet $(l, \alpha, \beta)$. We say that a vector-valued function $\Phi(\mathbf{w}) = (\Phi_1(\mathbf{w}), \ldots, \Phi_d(\mathbf{w}))$ is Pfaffian if each component $\Phi_i$ is Pfaffian with the same domain $U$. See Khovanskii [1991], Zell [1999], Gabrielov and Vorobjov [2004] for background and further details. The following shows that the set of Pfaffian functions is quite large.

**Theorem 7** (Khovanskii, "elementary functions are Pfaffian")**.** *Suppose that a function $g$ on a domain $U \subset \mathbb{R}^W$ is defined by a formula constructed from the variables $w_1, \ldots, w_W$ using finitely many real numbers, standard arithmetic operations $(+, -, \times, /)$, elementary functions $\ln$, $\exp$, $\sin$, $\arcsin$, and compositions. Suppose that for each $\mathbf{w} \in U$ the value $g(\mathbf{w})$ is well-defined in the sense that during the computation the functions $\ln$ and $\arcsin$ are applied on the intervals $(0, \infty)$ and $(-1, 1)$, respectively, and there is no division by 0. Moreover, suppose that there is a bounded interval $(a, b)$ to which the arguments of $\sin$ always belong for all $\mathbf{w} \in U$. Then the function $g$ is Pfaffian with complexity depending only on the size of the formula and the length of the interval $(a, b)$.*

Our theorem on learnable targets can then be stated as:

**Theorem 8** (B)**.** *Suppose that $\Phi : \mathbb{R}^W \to \mathbb{R}^d$ is a Pfaffian map and $W < d$. Then the closure $\overline{\Phi(\mathbb{R}^W)}$ has Lebesgue measure 0 in $\mathbb{R}^d$. In particular, the GF-learnable targets $F_\Phi$ have Lebesgue measure 0 in $\mathbb{R}^d$.*

If $\Phi$ is defined by elementary functions involving $\sin$ on an unbounded domain, then $\overline{\Phi(\mathbb{R}^W)}$ need not have Lebesgue measure 0. As the simplest family of examples, consider $\Phi = (\Phi_1, \ldots, \Phi_d)$ given

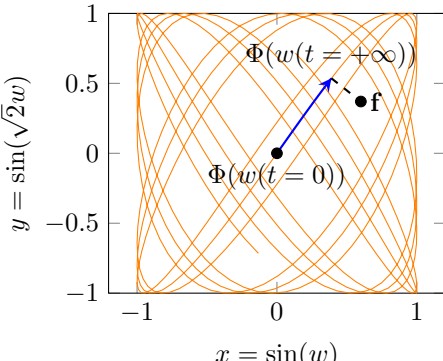

Figure 4: The curve $\Phi(w) = (\sin(w), \sin(\sqrt{2}w))$ densely fills the square $[-1, 1]^2$, but for all targets $\mathbf{f}$ except for a set of Lebesgue measure 0 the respective GF trajectory $w(t)$ is trapped at a spurious local minimum so that $\Phi(w(t)) \not\to \mathbf{f}$. Corollary 10 shows that this is true for all models (6) with any number of parameters $W < d$.

by

$$\Phi_i(\mathbf{w}) = \sin\Big(\sum_{j=1}^{W} a_{ij} w_j\Big), \quad i = 1, \dots, d, \tag{6}$$

with some constants $a_{ij}$. Kronecker's theorem [Kronecker, 1884, Gonek and Montgomery, 2016] implies that the points $(\sum_j a_{1j} w_j, \dots, \sum_j a_{dj} w_j)$ densely fill the torus $(\mathbb{R}/2\pi\mathbb{Z})^d$ as $\mathbf{w}$ runs over $\mathbb{R}^W$ whenever the vectors $\mathbf{a}_i = (a_{i1}, \dots, a_{iW}), i = 1, \dots, d$, are linearly independent over the rationals $\mathbb{Q}$. Accordingly, in this case $\overline{\Phi(\mathbb{R}^W)} = [-1, 1]^d$. However, the GF-learnable targets will still have Lebesgue measure 0 due to the prevalence of trapping local minima, as can be seen by a suitable extension of Theorem 3:

**Proposition 9** (C). *Let $1 \le W < d < \infty$ and $\Phi : \mathbb{R}^W \to \mathbb{R}^d$ be $C^2$. Suppose that for some open $U \subset \mathbb{R}^d$ the first and second derivatives of $\Phi$ are uniformly bounded on $\Phi^{-1}(U)$, and also the Jacobi matrix $J(\mathbf{w}) = \frac{\partial \Phi}{\partial \mathbf{w}}(\mathbf{w})$ is uniformly non-degenerate there in the sense that the lowest eigenvalue of $J^*(\mathbf{w})J(\mathbf{w})$ is uniformly bounded away from 0 on $\Phi^{-1}(U)$. Then $F_\Phi \cap U \subset \Phi(\mathbb{R}^W)$.*

**Corollary 10** (D). *Let $1 \le W < d < \infty$ and $\Phi : \mathbb{R}^W \to \mathcal{H} = \mathbb{R}^d$ be given by Eq. (6) with some constants $a_{ij}$. Then, regardless of these constants, the set $F_\Phi$ of respective learnable targets has Lebesgue measure 0.*

In Figure 4 we illustrate this result for $W = 1$.

Summarizing, Theorems 7, 8 imply that if the scalar components of the map $\Phi : \mathbb{R}^W \to \mathbb{R}^d$ with $W < d$ are elementary functions not involving $\sin$ (or related functions) acting on an unbounded domain, then $\Phi$ can GF-learn only exceptional targets in $\mathbb{R}^d$. Moreover, even if $\Phi$ involves $\sin$, there is no obvious mechanism how this would help GF-learn a non-negligible set of targets: examination of the basic family of $\sin$-models (6) in Corollary 10 suggests that GF trajectories would still be predominantly trapped at spurious minima.

## 6 Discussion

**Main takeaways.** Our results show that GF-learning with the number of parameters less than the target dimension is objectively problematic, but not impossible. By Theorems 3 and 4, the set of non-learnable targets is dense in the target space, while the set of learnable targets is not. Also, Theorem 8 shows that the set of learnable targets has zero Lebesgue measure for models expressible by elementary functions not involving $\sin$ on an unbounded domain.

Nevertheless, we have shown in Theorem 6 that if the targets are described by a known probability distribution, then it is possible to handcraft a (fairly complicated) model with just two parameters that learns the targets with probability arbitrarily close to 1. The learnable targets in our proof form a multi-dimensional Cantor set. Such a complicated structure is not surprising, since by Theorem 4 each subset of the target space homeomorphic to the 2-sphere must contain non-learnable targets. One can expect learnable sets to be more regular for models with a larger number of parameters (see an open question below).

**Open questions.** Our results leave various open questions, especially with regard to more detailed characterization of learnable sets of targets.

*Target measures with $\mu(\mathcal{H}) = \infty$.* It was crucial for our proof of main Theorem 6 that we could restrict our attenton to targets lying in a bounded box in $\mathbb{R}^d$. We could consider only such targets because if $\mu$ is a Borel measure on $\mathbb{R}^d$ and $\mu(\mathbb{R}^d) < \infty$, then $\mu(\mathcal{H} \setminus B)$ can be made arbitrarily small for a suitable bounded box $B$. However, one can ask if Theorem 6 also holds for measures with $\mu(\mathbb{R}^d) = \infty$, e.g. the Lebesgue measure. In this case there is no reduction to a bounded box, and our methods don't seem to work.

*Infinite-dimensional target spaces.* We prove Theorem 6 only for finite-dimensional target spaces $\mathcal{H}$, but one can also ask if it holds for an infinite-dimensional separable Hilbert space. As discussed in Section 2, this would cover the case of general distributions of inputs $\mathbf{x}$ for target functions.

*Non-density of the learnable targets for degenerate models.* Our proof that the subset $F_\Phi$ of learnable targets cannot be dense in the target space $\mathbb{R}^d$ if the number $W$ of parameters is less than $d$ (Theorem 3) heavily relies on the relatively strong assumption that $\Phi$ is $C^2$ and has a full rank Jacobian at the initial point. It would be interesting to clarify if this result holds without nondegeneracy assumptions and under weaker regularity assumptions, say for $\Phi$ differentiable with a Lipschitz gradient as sufficient for local integrability of the gradient flow.

*Learnable sets for general $1 < W < d$.* It would be interesting to generally describe target sets that can be GF-learned for $1 < W < d$. Our Theorem 6 only does that for $W = 2$ and for a family of multi-dimensional Cantor sets. Theorem 4 imposes weaker conditions on learnable target sets as $W$ increases (since higher-dimensional spheres contain lower-dimensional ones, but not the other way around), suggesting that with higher $W$ learnable sets become larger and more regular.

*Non-learnability using elementary functions involving* sin. As discusssed in Section 5, we expect that the high-probability learning proved in our main Theorem 6 cannot be established using models expressed through elementary functions, even if they involve sin with unbounded arguments.

## Acknowledgment

The author thanks the reviewers for several useful suggestions that helped improve the paper.

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

## A   Proof of Theorem 6

**Hierarchical structure of the learnable set.** We will construct a set $F_0$ such that $\mu(F_0) > 1 - \epsilon$ and for a suitable model $\Phi : \mathbb{R}^2 \to \mathbb{R}^d$ the targets $\mathbf{f}$ are $\Phi$-learnable, i.e. $F_0 \subset F_\Phi$, thus ensuring that $\mu(F_\Phi) > 1 - \epsilon$. We will occasionally refer to targets from $F_0$ (respectively, from the complement $\mathbb{R}^d \setminus F_0$) and the associated GF trajectories as *non-exceptional* (respectively, *exceptional*). The set $F_0$ has the form $F_0 = \cap_{n=1}^\infty \cup_\alpha B_\alpha^{(n)}$, where $B_\alpha^{(n)}$ is a nested hierarchy of rectangular boxes in $\mathbb{R}^d$ (see Figure 2). Each level-$n$ box $B_\alpha^{(n)}$ contains several non-intersecting level-$(n+1)$ sub-boxes $B_\beta^{(n+1)}$ of equal sizes (e.g. the big box shown on the right of Figure 3b contains 6 sub-boxes $B_\beta^{(n+1)}$).

The sub-boxes $B_\beta^{(n+1)}$ of the box $B_\alpha^{(n)}$ do not completely fill this box (we will need the gaps between them to support non-exceptional GF trajectories and to ensure their non-trapping in local minima; see again Figure 3b). We will ensure, however, that these gaps are small enough so that $\mu(F_0) > 1 - \epsilon$. In particular, we choose the root box $B = B^{(1)}$ as $B = [-c, c]^d$, where $c$ is large enough so that $\mu(B) > 1 - \epsilon/2$.

**Box splitting.** The next-level boxes $B_\beta^{(n+1)}$ are obtained from the parent box $B_\alpha^{(n)}$ by what we call *splitting* (also referred to as *carving* in the main text). Choose some sequences of *splitting coordinates* $x_{k_n}$ with $k_n \in \{1, \ldots, d\}$ and integer *splitting numbers* $s_n, n = 1, 2, \ldots$ We require that $k_{n+1} \neq k_n$ for all $n$.

The number of child boxes $B_\beta^{(n+1)}$ contained in the parent box $B_\alpha^{(n)}$ is $2s_n$ (e.g., $s_n = 3$ in Figure 3b). The splitting of the parent box $B_\alpha^{(n)}$ into $B_\beta^{(n+1)}$ only involves the coordinates $x_{k_n}$ and $x_{k_{n+1}}$. Specifically, if the parent box has $x_{k_n} \in [a, b]$, then the child boxes $B_\beta^{(n+1)}$ with $\beta = 1, \ldots, 2s_n$ have $x_{k_n} \in [a + (\beta - 1)h_n + \epsilon_n, a + \beta h_n - \epsilon_n]$, where $h_n = (b - a)/(2s_n)$ and $\epsilon_n > 0$ is a sufficiently small number. Regarding the other coordinate $x_{k_{n+1}}$, if the parent box has $x_{k_{n+1}} \in [c, d]$, then the child boxes will have $x_{k_{n+1}} \in [c + 2h_n, d]$ or $x_{k_{n+1}} \in [c, d - 2h_n]$, depending on whether the aligned piece of the level curve lies near the side with $x_{k_{n+1}} = c$ or with $x_{k_{n+1}} = d$ (see details on level curves below).

The splitting along $x_{k_n}$ will ensure that the GF trajectory gets closer to the target in the $(x_{k_n}, x_{k_{n+1}})$-plane, while the $x_{k_{n+1}}$-contraction of the boxes will be used to avoid getting trapped at a local minimum. We will return later to the conditions on the splitting numbers and coordinates necessary to ensure that $F_0 \subset F_\Phi$ and $\mu(F_0) > 1 - \epsilon$.

**The two parameters and $u$-monotonicity.** Let $u$ and $v$ be the two parameters of the model, so that $\mathbf{w} = (u, v)$. These two parameters will play very different roles. We will ensure that for all non-exceptional targets $\mathbf{f} \in F_0$, $u(t)$ is a monotone increasing function of $t$ on the whole GF trajectory ($u$-*monotonicity*). To this end, we ensure that the whole trajectory $\mathbf{w}(t)$ belongs to a domain in $\mathbb{R}^2$ in which $\partial_u L_\mathbf{f} < 0$, so that $\frac{du}{dt} > 0$ by definition of GF. In particular, this allows to parameterize the trajectories $\mathbf{w}(t)$ and $\Phi(\mathbf{w}(t))$ by $u \geq 0$.

**Level curves.** The map $\Phi$ can be described in terms of the curves $\Phi(u, v)$ where $u$ is fixed and $v$ varied. We refer to the respective straight lines $l_u = \{(u, v) : v \in \mathbb{R}\} \subset \mathbb{R}^2$ in the parameter space as *level lines* and the curves $\Phi(l_u) \subset \mathcal{H}$ in the target space as *level curves*. A GF trajectory $\mathbf{w}(t)$ can be specified by a single point on each level line $l_u$.

**Level-set-based description of $\Phi$.** It will be convenient to simplify the description of the map $\Phi$ by only describing the level curves $\Phi(l_u)$ as subsets of $\mathcal{H}$, without specifying parameterizations $v \mapsto \Phi(u, v)$. This simplified description can be justified by making the variable $u$ *slow* and keeping the variable $v$ *fast*, in the following sense. Suppose that we already have some map $\widetilde{\Phi} : \mathbb{R}^2 \to \mathcal{H}$, and define a new map $\Phi$ by stretching the variable $u$ by a large factor $\lambda$: $\Phi(u, v) = \widetilde{\Phi}(u/\lambda, v)$. This rescales the $u$-derivative: $\partial_u \Phi = \lambda^{-1} \partial_u \widetilde{\Phi}$. Then, at $\lambda \gg 1$ we have $|\partial_u \Phi| \ll |\partial_v \Phi|$ and accordingly $|\partial_u L_\mathbf{f}| \ll |\partial_v L_\mathbf{f}|$ unless $|\partial_v \Phi| \approx 0$. This means that GF associated with $\Phi$ proceeds in the $v$-direction much faster than in the $u$-direction (i.e., transition along a level curve occurs much faster than from one level curve to another) unless near a $v$-stationary point. This implies that each point $\mathbf{w}(t)$ of a GF trajectory can be approximately found by locally minimizing the distance to the target on the level curve $\Phi(l_{u(t)})$:

$$\mathbf{w}(t) \approx \mathbf{w}^*(u(t)); \quad \mathbf{w}^*(u) \overset{\text{def}}{=} \underset{\mathbf{w} \in l_u}{\arg\min} \|\mathbf{f} - \Phi(\mathbf{w})\|. \tag{7}$$

In general, the minimizer here should be local, but in our construction of $\Phi$ it will also be global. This approximation can be made arbitrarily accurate by sufficiently stretching the parameter $u$.

Since $\mathbf{w}^*(u)$ only depends on $u$, the trajectory $\mathbf{w}(t)$ is approximately independent of how level curves are parameterized by $v$. Accordingly, in the sequel we can ignore the details of this parameterization and just describe the level curves as subsets of $\mathcal{H}$ rather than functions of $v$.

**The aligned hierarchical structure of $\Phi$ (Fig. 3a).** Our construction of $\Phi$ is described in terms of a hierarchical structure of level sets aligned with the hierarchy of boxes $B_\alpha^{(n)}$. First, let $0 = u_0 < u_1 < u_2 < \ldots$ be a sequence of particular values of $u$. We construct the map $\Phi$ separately for each strip

$\{(u, v) : u_n \leq u \leq u_{n+1}\}$. We refer to the respective restriction $\Phi^{(n)} \equiv \Phi|_{u_n \leq u \leq u_{n+1}}$ as *the n'th stage* of $\Phi$. The monotonicity of $u(t)$ for targets $\mathbf{f} \in F_0$ will ensure that each respective trajectory $\mathbf{w}(t)$ sequentially passes through all the stages in their natural order. We denote by $l^{(n)} \equiv l_{u_n}$ the level lines serving as the boundaries for the domains of the stages $\Phi^{(n)}$.

Each stage $\Phi^{(n)}$ is associated with the splitting of the level-$n$ boxes $B_\alpha^{(n)}$ and can be viewed as defining a deformation of the level curve $\Phi(l^{(n)})$ to the level curve $\Phi(l^{(n+1)})$ lying closer to the targets in the new boxes $B_\beta^{(n+1)}$.

**Alignment between target boxes and level curves (Fig. 3b).** Each box $B_\alpha^{(n)}$ is accompanied by a respective *aligned piece* $\Phi(l_\alpha^{(n)})$ of the level curve $\Phi(l^{(n)})$, where $l_\alpha^{(n)}$ is a segment of the level line $l^{(n)}$, and the segments corresponding to different $\alpha$ are disjoint. The aligned piece $\Phi(l_\alpha^{(n)})$ is a straight line segment. It lies outside the box $B_\alpha^{(n)}$, but close to one of its 1D edges oriented along the splitting coordinate $x_{k_n}$. As $u$ increases from $u_n$ to $u_{n+1}$, the level curve segment $\Phi(l_\alpha^{(n)})$ is deformed in the new splitting direction $x_{k_{n+1}}$ so that $\Phi(l^{(n+1)})$ now contains segments $\Phi(l_\beta^{(n+1)})$ aligned with some $x_{k_{n+1}}$-oriented edges of the sub-boxes $B_\beta^{(n+1)}$, and the new stage $n + 1$ can commence.

We will ensure that each non-exceptional trajectory $\Phi(\mathbf{w}(t))$ goes through one of the aligned pieces $\Phi(l_\alpha^{(n)})$ at each stage $n$.

**Reformulations of $u$-monotonicity.** We need to ensure the $u$-monotonicity of non-exceptional trajectories, i.e. the condition $\partial_u L_{\mathbf{f}}(\mathbf{w}(t)) < 0$ holding on the whole trajectory $\mathbf{w}(t)$. In terms of the map $\Phi$, this condition reads

$$\langle \partial_u \Phi(\mathbf{w}(t)), \mathbf{f} - \Phi(\mathbf{w}(t)) \rangle > 0. \tag{8}$$

At a local minimizer $\mathbf{w}^*(u)$ given by Eq. (7) we have $\langle \partial_v \Phi(\mathbf{w}^*(u)), \mathbf{f} - \Phi(\mathbf{w}^*(u)) \rangle = 0$. Then, if a GF trajectory is approximated by the minimizer trajectory $\mathbf{w}^* = \mathbf{w}^*(u)$ (by stretching the parameter $u$), the condition of $u$-monotonicity becomes

$$\frac{d}{du} \|\mathbf{f} - \Phi(\mathbf{w}^*(u))\|^2 < 0. \tag{9}$$

One can also equivalently write this last condition in the form

$$\langle P_{\partial_v \Phi}^\perp \partial_u \Phi(\mathbf{w}^*(u)), \mathbf{f} - \Phi(\mathbf{w}^*(u)) \rangle > 0, \tag{10}$$

where $P_{\partial_v \Phi}^\perp \partial_u \Phi(\mathbf{w})$ denotes the projection of $\partial_u \Phi(\mathbf{w})$ to the direction orthogonal to $\partial_v \Phi(\mathbf{w})$ in the plane $(\partial_u \Phi(\mathbf{w}), \partial_v \Phi(\mathbf{w}))$. Condition (10) means geometrically that a level curve $\Phi(l_u)$ locally, near the point $\Phi(\mathbf{w}^*(u))$, gets closer to the target $\mathbf{f}$ as $u$ increases.

**Transition from $\Phi(l^{(n)})$ to $\Phi(l^{(n+1)})$ (Fig. 3c).** During the $n$'th stage, all the components of the map $\Phi^{(n)} : \mathbb{R}^2 \to \mathbb{R}^d$ are constant in the strip $\{(u, v) : u_n + \epsilon < u < u_{n+1} - \epsilon\}$ except for the components $k_n$ and $k_{n+1}$ associated with splitting directions. Here, we consider a substrip $u_n + \epsilon < u < u_{n+1} - \epsilon$ of the full stage-$n$ strip $u_n \leq u \leq u_{n+1}$ because we need to ensure that $\Phi$ is $C^\infty$. In the remaining narrow substrips $u_n \leq u \leq u_n + \epsilon$ and $u_{n+1} - \epsilon \leq u \leq u_{n+1}$ the map $\Phi^{(n)}$ is smoothly connected to the maps $\Phi^{(n-1)}$ and $\Phi^{(n+1)}$, respectively. It is easy to see that such a smooth connection can be ensured with arbitrarily small $\epsilon$ by additionally suitably varying the components $k_{n-1}$ and $k_{n+1}$ in the respective substrips; we omit these details.

We construct the map $\Phi$ so that each aligned piece $\Phi(l_\alpha^{(n)})$ is a segment of the straight line oriented in the splitting direction $x_{k_n}$. In each box $B_\alpha^{(n)}$, the transformation of the level curve to the $2s_n$ next-level pieces is performed symmetrically (Fig. 3b), so we need to only describe the transformation in one of these $2s_n$ sub-boxes (Fig. 3c).

Suppose for simplicity that the sub-box $B_\alpha^{(n)}$ in question has the form $[0, a] \times [0, b]$ w.r.t. the coordinates $(x_{k_n}, x_{k_{n+1}})$ and the aligned curve $\Phi(l^{(n)})$ lies to the left as in Fig. 3c (a general case is treated similarly using a translation and possibly a reflection). The intermediate level curves $\Phi(l_u), u_n \leq u \leq u_{n+1}$, can then be generally described in the $(x_{k_n}, x_{k_{n+1}})$ plane as formed in two sub-stages separated by some $u^* \in (u_n, u_{n+1})$.

1. **"Gathering" sub-stage** $u \in [u_n, u^*]$: for some $\epsilon > 0$ the level curve $\Phi(l_u)$ can be given on the segment $x_{k_n} \in [\epsilon, a - \epsilon]$ by $x_{k_{n+1}} = \alpha(u)x_{k_n} - \widetilde{\epsilon}(u)$ with some small monotone decreasing $\widetilde{\epsilon}(u) > 0$ and some monotone increasing $\alpha(u)$ such that $\alpha(u_n) = 0$ and $\alpha(u^*) = 1$. In particular, at $u = u^*$ the level curve is given by $x_{k_{n+1}} = x_{k_n} - \widetilde{\epsilon}(u^*)$. In the remaining small segments $x_{k_n} \in [0, \epsilon] \cup [a - \epsilon, a]$ neighboring with the other sub-boxes the level curve is extended in some way (by suitable arcs) to ensure its smoothness and, moreover, concavity on the interval $(a - \epsilon, a]$ as a function $x_{k_{n+1}} = x_{k_{n+1}}(x_{k_n})$.

2. **"Spreading" sub-stage** $u \in [u^*, u_{n+1}]$: the level curve $\Phi(l_u)$ is evolved by extending its tip at $x_{k_n} = a$ all the way to $x_{k_{n+1}} > b$. Specifically, $\Phi(l_u)$ contains a straight line segment $I(u)$ parallel to the axis $x_{k_{n+1}}$ and eventually, at $u = u_{n+1}$, including the aligned piece $\Phi(l_\beta^{(n+1)})$. The $x_{k_n}$ position of the segment $I(u)$ is monotone decreasing in $u$, but remains close to $a$. The $x_{k_{n+1}}$ position of the left endpoint of $I(u)$ remains close to $a$, while for the right endpoint $x_{k_{n+1}}$ increases from around $a$ to $b$ as $u$ increases from $u^*$ to $u_{n+1}$. The right endpoint remains smoothly connected by a suitable concave arc to the analogous point in the neighboring box. See Fig. 3c for an illustration.

The idea of this whole construction is to ensure that, as $u$ is increased, the points on the level curves $\Phi(l_u)$ closest to the target $\mathbf{f}$ go around the corner of the box $B_\beta^{(n+1)}$ as shown in Fig. 3c. The purpose of the "gathering" sub-stage is to force the trajectory $\Phi(\mathbf{w}(u))$ by $u = u^*$ to get to the tip of the level curve $\Phi^{(n)}(l_{u^*})$ at $x_{k_n} \approx a$. Then, in the "spreading" sub-stage the trajectory remains close to the moving tip until its coordinate $x_{k_{n+1}}$ reaches the respective coordinate $f_{k_{n+1}}$ of the target $\mathbf{f}$, after which the trajectory slips off the tip to the straight line segment $I(u)$ (the concavity of the arcs mentioned above ensures that the trajectory is not trapped on the arcs). The trajectory then maintains the target coordinate $(\Phi(\mathbf{w}(u)))_{k_{n+1}} = f_{k_{n+1}}$ until reaching the aligned piece $\Phi(l_\beta^{(n+1)})$.

Let us discuss now how the above picture may break down for some targets $\mathbf{f}$. First, it breaks down if the pair of target components $(f_{k_n}, f_{k_{n+1}})$ belongs to the domain swept by $\Phi^{(n)}$ (i.e., $(f_{k_n}, f_{k_{n+1}}) \in \Phi^{(n)}(l_{\widetilde{u}})$ for some $\widetilde{u} \in (u_n, u_{n+1})$). In this case the $u$-monotonicity conditions (9), (10) are violated for $u \geq \widetilde{u}$ and the GF trajectory gets trapped at a local minimum.

Next, for some $\mathbf{f}$ the $u$-monotonicity holds, but the trajectory $\Phi(w)$ fails to reach the tip region $x_{k_n} \approx a$. This occurs for the targets $\mathbf{f}$ such that $f_{k_n} + f_{k_{n+1}} \leq 2a$ – for such targets the trajectory gets stuck near the orthogonal projection of $(f_{k_n}, f_{k_{n+1}})$ to the line $x_{k_{n+1}} = x_{k_n}$.

Finally, if $f_{k_n} = 0$, then the $k_n$-component of the trajectory is stuck at 0 too. Moreover, if $f_{k_n}$ is nonzero but close to 0, then the trajectory $\Phi(\mathbf{w}(u)))$ remains close to the plane $x_{k_n} = 0$ and so again fails to reach the tip region. (This happens because near this plane $\partial \Phi(\mathbf{w}(u_n)) \approx 0$, invalidating our argument that we can ensure $|\partial_u L_{\mathbf{f}}| \ll |\partial_v L_{\mathbf{f}}|$ by stretching the variable $u$ – the necessary stretching blows up as $f_{k_n} \to 0$.)

If the target $\mathbf{f}$ lies outside these three regions, then the map $\Phi^{(n)}$ succeeds in guiding the trajectory $\Phi(\mathbf{w}(t))$ from a point on the aligned piece $\Phi(l_\alpha^{(n)})$ near the orthogonal projection of $\mathbf{f}$ of that piece to a point on the aligned piece $\Phi(l_\beta^{(n+1)})$ near the orthogonal projection of $\mathbf{f}$ to this piece. In particular, it is sufficient to require that $(f_{k_n}, f_{k_{n+1}})$ belongs to the rectangle $R = [\epsilon, a - \epsilon] \times [2a, b]$ with some $\epsilon > 0$: we can then suitably stretch $u$ and adjust the map $\Phi^{(n)}$ so that for all targets with $(f_{k_n}, f_{k_{n+1}}) \in R$ the above transition holds, and moreover with desired accuracy uniform in $R$.

**The initial map $\Phi^{(0)}$.** The construction of the initial map $\Phi^{(0)}$ is slightly different (and simpler) than the above inductive construction for general stage $n$. The GF starts from the particular point $\mathbf{w} = 0$ of the left level line $l^{(0)} = \{(u, v) : u = 0\}$ of stage 0. Its right level line $l^{(1)} = \{(u, v) : u = u_1\}$ has a single aligned piece $\Phi(l_1^{(1)})$ aligned with the initial box $B^{(1)}$ at some 1D edge along the direction $x_{k_1}$. We only need to ensure that for all targets $\mathbf{f}$ in the initial box $B^{(1)}$, the GF trajectory approaches by $u = u_1$ a point on $\Phi(l_1^{(1)})$ close to the orthogonal projection of $\mathbf{f}$ to $\Phi(l_1^{(1)})$.

Recall that $B^{(1)} = [-c, c]^d$. Suppose, for example, that the edge in question is $[c, -c] \times (c, c, \ldots, c)$ and the aligned piece of level line is $[c, -c] \times (a, a, \ldots, a)$ with some $a > c$. Then we can define

$\Phi^{(0)}$ as the linear map

$$\Phi_i^{(0)}(u, v) = \begin{cases} v, & i = 1 \\ a + 1 - \frac{u}{u_1}, & i > 1. \end{cases} \qquad (11)$$

It is easy to see that for all $\mathbf{f} \in B^{(1)}$ the respective GF trajectory $(u(t), v(t))$ satisfies $\frac{du}{dt} > 0$ and, by choosing $u_1$ large enough, at $t_1$ such that $u(t_1) = u_1$ we will have $|v(t) - f_1| < \epsilon$ with arbitrarily small $\epsilon$.

**Ensuring convergence** $\inf_t L_\mathbf{f}(\mathbf{w}(t)) = 0$ **for all** $\mathbf{f} \in F_0$. The presented construction of the maps $\Phi^{(n)}$ and the boxes $B_\beta^{(n+1)}$ ensures that for all targets $\mathbf{f} \in B_\beta^{(n+1)}$ the following approximately occurs with the components of the respective discrepancy vector $\mathbf{f} - \Phi(\mathbf{w}(t))$ as a result of the $n$'th stage of GF:

1. The component $k_{n+1}$ approximately vanishes.

2. The component $k_n$, which is approximately zero at the beginning of the stage, becomes nonzero, but limited by the size of $B_\beta^{(n+1)}$ in the $k_n$'th coordinate direction.

3. The other components remain approximately the same.

("Approximately" here means minor corrections due to the imperfect approximation by level curve minimizers (7), due to gaps between the boxes and the aligned level curves, and due to smoothing of the overall map $\Phi$ at the boundaries of the restrictions $\Phi^{(n)}$; all these corrections can be made arbitrarily small).

Clearly, it follows that if the sequence $k_n$ takes each of the values $1, \ldots, d$ infinitely many times, then, since the size of $B_\beta^{(n+1)}$ in any direction vanishes in the limit $n \to \infty$, the vectors $\mathbf{f} - \Phi(\mathbf{w}(t))$ converge to $\mathbf{0}$ for all targets from $F_0 = \cap_{n=1}^\infty \cup_\alpha B_\alpha^{(n)}$.

**Ensuring** $\mu(F_0) \geq 1 - \epsilon$. The set $F_0$ is obtained by removing a countable number of rectangular parts from the root box $B^{(1)}$ that was chosen to have measure $\mu(B^{(1)}) > 1 - \epsilon/2$. Accordingly, we only need to show that the removed part can be made arbitrarily small with respect to the measure $\mu$. The removed parts are of two kinds (see Fig. 3b):

1. Those associated with the gaps between sub-boxes $B_\beta^{(n+1)}$ in the direction $k_n$.

2. Those associated with the gaps between the aligned pieces $\Phi(l)$ and sub-boxes $B_\beta^{(n+1)}$ in the direction $k_{n+1}$.

The width of the parts of the first kind can be made arbitrarily small. The width of the parts of the second kind was shown to be approximately equal to $2a$, where $a$ is the width of the sub-box $B_\beta^{(n+1)}$ in the direction $k_n$ (see Fig. 3c). However, $a$ can also be made arbitrarily small by increasing the splitting number $s_n$.

Employing the $\sigma$-additivity of $\mu$, we conclude that the only obstacle to making the measure of the removed parts arbitrarily small is that if some of the splitting hyperplanes of co-dimension 1 used in the construction have positive measure (and so we cannot make the measure of the removed parts arbitrarily small by decreasing their width). However, the number of values $c$ such that $\mu(\{x_k = c\}) > 0$ is countable, so we can avoid such hyperplanes by slightly shifting the root box $B^{(1)}$.

## B Proof of Theorem 8

The crucial property of Pfaffian functions, due to Khovanskii, is that their level sets can only have a bounded number of connected components. This result relies on some mild assumptions on the domain $U$; we will assume for simplicity that $U = \mathbb{R}^W$ (see Remark 2.12 in Gabrielov and Vorobjov [2004]). We state the result in a form suitable for our purposes (see, e.g., Corollary 3.3 in Gabrielov and Vorobjov [2004]).

**Theorem 11.** *Let $g_1, \ldots, g_k$ be Pfaffian functions on $\mathbb{R}^W$. Then the number of connected components of the set*

$$\{\mathbf{w} \in \mathbb{R}^W : g_1(\mathbf{w}) = \ldots = g_k(\mathbf{w}) = 0\} \tag{12}$$

*is bounded by a finite value only depending on $k$ and the complexities of the functions $g_i$.*

An important special case occurs if $k = W$ and the solutions of the system (12) are non-degenerate in the sense that the respective Jacobians $\frac{\partial g_i}{\partial w_j}(\mathbf{w})$ are non-degenerate. In this case the level set (12) consists of isolated points, and their number is bounded by a number only depending on the complexities of the functions $g_i$.

The proof of Theorem 8 is a reduction to Theorem 11.

*Proof of Theorem 8.* It is sufficient to consider the case $W = d - 1$ (by trivially extending $\Phi$ to more arguments). It is also sufficient to prove that $\overline{\Phi(\mathbb{R}^W)}$ has Lebesgue measure 0 in the cube $[0,1]^d$: by rescaling and shifting, it then has Lebesgue measure 0 in any other cube and then, by $\sigma$-additivity, in the whole space $\mathbb{R}^d$.

Let $N$ be a large integer. Consider the $d$-dimensional grid of cubes $\Delta_{\mathbf{n}}$ of size $\frac{1}{N}$ given by

$$\Delta_{\mathbf{n}} = \{\mathbf{y} \in \mathbb{R}^d : y_i^* + \tfrac{n_i}{N} \leq y_i \leq y_i^* + \tfrac{n_i+1}{N}\}, \quad \mathbf{n} = (n_1, \ldots, n_d) \in \mathbb{Z}^d, \tag{13}$$

where $\mathbf{y}^* = (y_1^*, \ldots, y_W^*)$ is some reference point.

For each $i = 1, \ldots, d$ consider the map $\widehat{\Phi}_i : \mathbb{R}^W \to \mathbb{R}^{d-1} \cong \mathbb{R}^W$ obtained from $\Phi$ by removing the $i$'th component, i.e. $\widehat{\Phi}_i = (\Phi_1, \ldots, \Phi_{i-1}, \Phi_{i+1}, \ldots, \Phi_d)$. We will choose $\mathbf{y}^*$ so that for each $i$ and $\widehat{\mathbf{n}} \in \mathbb{Z}^W$ the point

$$\mathbf{y}_{i,\widehat{\mathbf{n}}} = (y_1^*, \ldots, y_{i-1}^*, y_{i+1}^*, \ldots, y_d^*) + \tfrac{\widehat{\mathbf{n}}}{N} \tag{14}$$

is a non-critical value of $\widehat{\Phi}_i$, i.e. for any $\mathbf{w}$ such that $\widehat{\Phi}_i(\mathbf{w}) = \mathbf{y}_{i,\widehat{\mathbf{n}}}$ the Jacobian $\frac{\partial \widehat{\Phi}_i}{\partial w_j}(\mathbf{w})$ is non-degenerate. To this end, recall that by Sard's theorem the set $V_i$ of critical values of $\widehat{\Phi}_i$ has Lebesgue measure 0. Then the set $V_i' = \cup_{\widehat{\mathbf{n}} \in \mathbb{Z}^W}(V_i - \frac{\widehat{\mathbf{n}}}{N})$ also has Lebesgue measure 0 in $\mathbb{R}^{d-1}$. It follows that $V_i'' = \{\mathbf{y} \in \mathbb{R}^d : (y_1, \ldots, y_{i-1}, y_{i+1}, \ldots, y_d) \in V_i'\}$ has Lebesgue measure 0 in $\mathbb{R}^d$. Then $V = \cup_{i=1}^d V_i''$ also has Lebesgue measure 0 in $\mathbb{R}^d$. The complement $\mathbb{R}^d \setminus V$ is precisely the set of all $\mathbf{y}^*$ such that the values (14) are non-critical for all $i$ and $\widehat{\mathbf{n}}$. Since $\mathbb{R}^d \setminus V$ has full Lebesgue measure, we can find a suitable $\mathbf{y}^*$; moreover, we can choose it so that $0 < y_i^* < \frac{1}{N}$, which will be convenient.

Consider the cubes $\Delta_{\mathbf{n}}$ lying in $[0,1]^d$, i.e., with $\mathbf{n} \in \{0, \ldots, N-2\}^d$. There are $(N-1)^d$ such cubes. Suppose that the interior of a cube $\Delta_{\mathbf{n}}$ contains a point of $\overline{\Phi(\mathbb{R}^W)}$. Then, since $\Phi(\mathbb{R}^W)$ is connected, either $\Phi(\mathbb{R}^W) \subset \Delta_{\mathbf{n}}$ or $\Phi(\mathbb{R}^W)$ intersects the boundary of $\Delta_{\mathbf{n}}$. In the first case the Lebesgue measure of $\overline{\Phi(\mathbb{R}^W)}$ does not exceed $N^{-d}$, so if this case occurs for arbitrarily large $N$, $\overline{\Phi(\mathbb{R}^W)}$ has Lebesgue measure 0.

We can thus assume that if the interior of $\Delta_{\mathbf{n}}$ contains a point of $\overline{\Phi(\mathbb{R}^W)}$, then the boundary of $\Delta_{\mathbf{n}}$ intersects $\Phi(\mathbb{R}^W)$. We will argue now that the number of such cubes $\Delta_{\mathbf{n}}$ in $[0,1]^d$ is $O(N^{d-1})$, i.e. is vanishing compared to the total number $(N-1)^d$ of the cubes.

Indeed, the boundary of $\Delta_{\mathbf{n}}$ consists of cubic faces of dimensions $1, \ldots, d-1$ (ignoring the vertices, i.e. faces of dimension 0). Let $s$ be the lowest dimension of a face intersecting $\Phi(\mathbb{R}^W)$. Denote by $R$ such a face. Consider two cases.

1. $s = 1$. In this case the face $R$ is a segment oriented along some coordinate $i$ and with the other coordinates forming a point $\mathbf{y}_{i,\widehat{\mathbf{n}}}$ of the form (14). Recall that by construction the points $\mathbf{y}_{i,\widehat{\mathbf{n}}}$ are non-critical values of the map $\widehat{\Phi}_i$. The pre-image $\Phi^{-1}(R)$ is a subset of $\widehat{\Phi}_i^{-1}(\mathbf{y}_{i,\widehat{\mathbf{n}}})$ and so it consists of isolated points. Since $\Delta_{\mathbf{n}} \subset [0,1]^d$, the multi-index $\widehat{\mathbf{n}}$ of the segment belongs to $\{0, \ldots, n-1\}^{d-1}$. By Theorem 11, the number of points in $\widehat{\Phi}_i^{-1}(\mathbf{y}_{i,\widehat{\mathbf{n}}})$ is bounded by a constant only depending on the complexity of the map $\Phi$. The total number of points in the pre-image $\widehat{\Phi}_i^{-1}(\{\mathbf{y}_{i,\widehat{\mathbf{n}}} : \widehat{n} \in \{0, \ldots, n-1\}^{d-1}\})$ is then $O(N^{d-1})$. It follows that the total number of points mapped by $\Phi$ to one-dimensional faces of all the

cubes $\Delta_\mathbf{n} \subset [0,1]^d$ is $O(N^{d-1})$. Since each of these faces is a face for at most $2^d$ cubes, we conclude that the total number of cubes $\Delta_\mathbf{n} \subset [0,1]^d$ whose one-dimensional faces intersect $\Phi(\mathbb{R}^W)$ is $O(N^{d-1})$.

2. $s > 1$. In this case the intersection $R \cap \Phi(\mathbb{R}^d)$ lies in the interior of the face $R$. Let $I = \{i_1, \ldots, i_s\}$ be the coordinates along which the face is oriented. Similarly to the case $s = 1$, consider the map $\widehat{\Phi}_I$ obtained from $\Phi$ by dropping all the components $i \in I$. The pre-image $\Phi^{-1}(R)$ lies in the pre-image $\widehat{\Phi}_I^{-1}(\mathbf{y})$ of the point $\mathbf{y} \in \mathbb{R}^{d-s}$ formed by the coordinates $i \notin I$ of the face $R$. Since $R \cap \Phi(\mathbb{R}^d)$ lies in the interior of the face $R$, the pre-image $\Phi^{-1}(R)$ is a subset of $\widehat{\Phi}_I^{-1}(\mathbf{y})$ disconnected from the rest of $\widehat{\Phi}_I^{-1}(\mathbf{y})$. In particular, if there are several $s$-dimensional faces (of several cubes) with the coordinates $i \notin I$ forming the same point $\mathbf{y}$ and having non-empty intersections with $\Phi(\mathbb{R}^d)$ that also lie in their interiors, then there are at least as many connected components in the pre-image $\widehat{\Phi}_I^{-1}(\mathbf{y})$. By Theorem 11, the number of these connected components is bounded by a constant. Since possible values $\mathbf{y}$ belong to a $(d-s)$-dimensional grid with spacing $\frac{1}{N}$, the total number of such faces in the cube $[0,1]^d$ (over all possible $\mathbf{y}$'s) is $O(N^{d-s})$, and then the number of respective cubes is also $O(N^{d-s})$.

Taking the limit $N \to \infty$, we conclude that the Lebesgue measure of $\overline{\Phi(\mathbb{R}^W)}$ is 0 as desired. $\quad\square$

## C   Proof of Proposition 9

Let $\mathbf{f} \in U$ and $L_\mathbf{f}(\mathbf{w}_0) < \epsilon$ for some $\mathbf{w}_0$. We will show that if $\epsilon$ is small enough, then there is a barrier of high loss at the sphere bounding the ball $B_{\mathbf{w}_0, r} = \{\mathbf{w} \in \mathbb{R}^W : \|\mathbf{w} - \mathbf{w}_0\| = r\}$ with a suitable radius $r$. Then, $\mathbf{f}$ belongs to the closure $\overline{\Phi(B_{\mathbf{w}_0, r})}$, while, by compactness and continuity, $\overline{\Phi(B_{\mathbf{w}_0, r})} = \Phi(B_{\mathbf{w}_0, r}) \subset \Phi(\mathbb{R}^W)$.

Assuming $\epsilon$ and $\Delta\mathbf{w} \equiv \mathbf{w} - \mathbf{w}_0$ are sufficiently small so that the segment $[\Phi(\mathbf{w}_0), \Phi(\mathbf{w})] \subset \Phi^{-1}(U)$, we have $c\|\Delta\mathbf{w}\| \le \|J(\mathbf{w}_0)\Delta\mathbf{w}\| \le C\|\Delta\mathbf{w}\|$ and $\|\Phi(\mathbf{w}) - \Phi(\mathbf{w}_0) - J(\mathbf{w}_0)\Delta\mathbf{w}\| \le C\|\Delta\mathbf{w}\|^2$ with some $U$-dependent constant $0 < c, C < \infty$. Then, with $\|\Delta\mathbf{w}\| = r$,

$$L_\mathbf{f}(\mathbf{w}) - L_\mathbf{f}(\mathbf{w}_0) = \tfrac{1}{2}\|\mathbf{f} - \Phi(\mathbf{w})\|^2 - \tfrac{1}{2}\|\mathbf{f} - \Phi(\mathbf{w}_0)\|^2 \tag{15}$$

$$= \tfrac{1}{2}\|\Phi(\mathbf{w}) - \Phi(\mathbf{w}_0)\|^2 - \langle \mathbf{f} - \Phi(\mathbf{w}_0), \Phi(\mathbf{w}) - \Phi(\mathbf{w}_0)\rangle \tag{16}$$

$$\ge \tfrac{1}{2}\|J(\mathbf{w}_0)\Delta\mathbf{w}\|^2 - C\|J(\mathbf{w}_0)\Delta\mathbf{w}\|\|\Delta\mathbf{w}\|^2 - \sqrt{2\epsilon}(\|J(\mathbf{w}_0)\Delta\mathbf{w}\| + C\|\Delta\mathbf{w}\|^2) \tag{17}$$

$$\ge \tfrac{c^2}{2}r^2 - C^2 r^3 - \sqrt{2\epsilon}C(r + r^2). \tag{18}$$

Choosing $r = \epsilon^{1/3}$, at small $\epsilon$ we get $L_\mathbf{f}(\mathbf{w}) - L_\mathbf{f}(\mathbf{w}_0) > 0$ uniformly on $\partial B_{\mathbf{w}_0, r}$, as desired.

## D   Proof of Corollary 10

First, observe that the loss is constant along directions orthogonal to the rows of the matrix $A = (a_{ij})$, and GF is orthogonal to these directions. Therefore, by performing an orthogonal transformation and discarding these directions, we can assume without loss of generality that $\Phi_i(\mathbf{w}) = \sin(\sum_{j=1}^W a_{ij}w_j + b_i)$ with some constants $b_i$ and a matrix $A$ that has full rank $W$. The Jacobian $J(\mathbf{w}) = \frac{\partial\Phi}{\partial\mathbf{w}}(\mathbf{w})$ can be represented as $J(\mathbf{w}) = D(\mathbf{w})A$, where $D(\mathbf{w}) = \text{diag}[\cos(\sum_{j=1}^W a_{ij}w_j + b_i), i = 1, \ldots, d]$.

Now let $U = (-c, c)^d$ with some $0 < c < 1$. Then, for all $\mathbf{w} \in \Phi^{-1}(U)$, the diagonal elements $\cos(\sum_{j=1}^W a_{ij}w_j + b_i)$ of the matrix $D(\mathbf{w})$ are uniformly separated from 0 by a distance not less than $\sqrt{1 - c^2} > 0$. Hence, in the operator sense, $J^*(\mathbf{w})J(\mathbf{w}) = A^*D^2(\mathbf{w})A \ge (1 - c^2)A^*A \ge \widetilde{c}\mathbf{1}$ for some $\widetilde{c} > 0$, i.e. $J(\mathbf{w})$ is uniformly non-degenerate on $\Phi^{-1}(U)$. Applying Proposition 9, we conclude that $F_\Phi \subset \partial[-1, 1]^d \cup \Phi(\mathbb{R}^W)$, which has Lebesgue measure 0 in $\mathbb{R}^d$.

