# OpenReview forum: "Learnability of high-dimensional targets by two-parameter models and gradient flow"
_NeurIPS.cc/2024/Conference — NeurIPS 2024 poster_

### Official Review · Reviewer_qWpD · 2024-07-11

**Soundness:** 3
**Presentation:** 2
**Contribution:** 1
**Rating:** 4
**Confidence:** 2

**Summary:**

This paper studies the problem of learning a target function $f$ lying in some $d$-dimensional Hilbert space $\mathcal{H}$ via gradient flow on a $W$-parameter model with $W < d$. The main result, Theorem 5, is that for $W = 2$, given a distribution over $\mathcal{H}$ there exists a parametric map $\Phi : \mathbb{R}^2 \rightarrow \mathcal{H}$ such that gradient flow converges to the target with probability approaching 1. In contrast, if the map $\Phi$ is constructed via "elementary functions," then the set of learnable targets has Lebesgue measure 0.

**Strengths:**

- The results of the novel and quite technically impressive, though I admit I am not so familiar with the related literature and did not fully follow the proof sketch.
- For the most part the paper is well written, and I do like the addition of Figure 2 to help understand the proof of the main theorem.

**Weaknesses:**

- I struggle to understand the relevance of the problem studied in this paper to the NeurIPS community / the field of Machine Learning. The construction for $\Phi$ in Theorem 5 is quite pathological, whereas parametric models used in ML/statistics are much more regular. In fact, in Section 5 the authors prove that if $\Phi$ is an elementary function, then only a measure 0 set of targets are learnable by GD. I thus do not see the significance of studying parametric models $\Phi$ which essentially act as space-filling curves and map $\mathbb{R}^2$ to $\mathbb{R}^d$ in a very complicated manner.

- Furthermore, in ML settings, when the parametric map $\Phi$ cannot express the ground truth target $f$, then the goal is to instead converge to the best possible predictor i.e obtain a loss $\inf_w \frac12\||f - \Phi(w)\||^2_{\mathcal{H}}$. Prior works studying the loss landscape of underparametrized models typically operate in the setting where the number of parameters is fewer than the number of *data points* and thus the training loss cannot go to zero. This paper is substantially different, and focuses on the problem where the number of parameters is smaller than the dimension of the function class, but we are still interested in learning a large portion of this function class, which to me feels quite strange.

- I find the proof sketch for Theorem 5, in particular a description of the construction of $\Phi$, to be unclear. A more detailed definition of objects like the Cantor set $F_0$ is needed, since this may not be familiar to the NeurIPS optimization/theory community. I also don't think the boxes $B_\alpha^{(n)}$ are ever defined in the main text.

**Questions:**

- Can you provide intuition on why Theorem 5 does not apply to infinite-dimensional target spaces?

---

> ### Author Rebuttal · Authors · 2024-08-07
>
> Thank you for the careful reading of our paper and your useful feedback and critique.
>
> * "*I struggle to understand the relevance of the problem studied in this paper to the NeurIPS community.. The construction for $\Phi$ in Theorem 5 is quite pathological..*"
>
> That's true, the construction in the proof looks pathological. But suppose you know nothing about the proof and just see the statement of Theorem 5. Would you say that this statement is irrelevant to ML/NeurIPS? We believe that it involves only natural and standard ML concepts (gradient descent, parametric models, target spaces ...), the only nonstandard thing being the scenario of a small number of parameters. In our opinion, the question addressed by Theorem 5 is not unreasonable and certainly not trivial. Our goal was to understand what can theoretically happen in this scenario, be it with or without complicated constructions. If we could prove Theorem 5 without complicated constructions, we would have done so.
>
> Another point to mention, Theorem 5 deals with the extreme case of just two parameters and a full convergence to the targets. We expect (and briefly mention in the Discussion) the pathologies to weaken if, e.g., the number of parameters is closer to the dimensionality of the target space.
>
> Also, while the parametric models used in ML/statistics are indeed more regular, modern neural networks can still be very complex in terms of their architectures, activation functions, etc. Their learning dynamics is not well-understood. We don't see why Theorem 5 could not potentially be relevant for some of their learning poperties (though this is of course purely hypothetical; we don't claim at this point any specific direct connection to practical models).
>
> * "*..the number of parameters is smaller than the dimension of the function class, but we are still interested in learning a large portion of this function class, which to me feels quite strange.*"
>
> But we also know that modern neural networks often contain much more parameters that the number of training data points. The dimensionality of the function class in this case is certainly much smaller than the number of parameters. If this is a valid scenario, why is it unreasonable to theoretically examine the opposite scenario?
>
> * "*I find the proof sketch for Theorem 5 .. to be unclear*"
>
> Thank you for this feedback. We admit that some elements of the proof sketch may be not clear enough. We wanted to convey the key ideas, but due to the size constraints we found it hard to give more than a flavor of the proof. We'll try to improve this sketch. Meanwhile, please see Appendix A, where the full proof is carefully described.

---

> > ### Comment · Reviewer_qWpD · 2024-08-09
> > **Response**
> >
> > Thank you to the authors for their response.
> >
> > I still find that the problem studied in this paper is not well motivated. It seems to me that in order for a function to be "GF-learnable" per the definition in the paper, the image of $\Phi(w)$ must cover most of $\mathcal{H}$ and thus the map $\Phi$ must necessarily be very pathological; no reasonable statistical model will be of this form. In fact, I do not know of any examples in the statistics/ML literature where one is interested in *exactly* fitting a dimension $d$ model with $W < d$ parameters. The usual setup in the underparameterized setting is not to obtain exactly zero training loss, but rather to converge to the predictor with the smallest loss over the candidate class of functions. While modern neural networks are indeed very complicated, I am very skeptical that they behave like the construction $\Phi$ presented in this work. And while I acknowledge that theoretical work does not have to have a direct practical impact, I do believe that theoretical work should model/explain some relevant aspect of reality, which I don't believe is accomplished by this paper. I thus am leaning towards maintaining my original score.

---

> > > ### Author Response · Authors · 2024-08-11
> > >
> > > Thank you for this feedback. We understand and appreciate your view, but respectfully would like to challenge it.
> > >
> > > > I do believe that theoretical work should model/explain some relevant aspect of reality, which I don't believe is accomplished by this paper.
> > >
> > > **Comparison with KST.** Let us compare our main theorem 5 with Kolmogorov Superposition Theorem (KST) saying that multivariate continuous functions can be represented in terms of univariate ones and summations. KST has inspired many papers/ideas/discussions related to ML/neural networks, primarily with the focus on confirming or increasing expressiveness of ML models. One work, very recent but already having received much attention, is mentioned by Reviewer MmYK [1]. Some others are cited in our paper.
> > >
> > > The construction used in KST is arguably much more pathological than ours in Theorem 5. The functions used in KST are very complicated continuous functions; they cannot be chosen as elementary or smooth, even locally. Also, the representation of the target function is obtaned by a fairly non-constructive argument.
> > >
> > > In contrast, our map $\Phi$ is smooth, can be chosen piecewise elementary, and the fitting is performed by standard gradient flow.
> > >
> > > Accordingly, we don't see why our work models/explains relevant aspects of reality worse than KST. If ML community finds KST useful, why should it not find useful our work, in which the model is much more regular and the fitting procedure the most standard?
> > >
> > > We also emphasize that while ideas of using KST for high expressiveness of ML models have a long history dating back to Hecht-Nielsen [2] and Kůrková [3], we are not aware of any previous works analyzing to which extent high expressiveness can be combined with gradient flow. We believe our work to be the first in this respect, and moreover providing a fairly balanced and comprehensive analysis.
> > >
> > > [1] KAN: Kolmogorov–Arnold Networks. Liu et al. 2024. arXiv:2404.19756.
> > > [2] R. Hecht-Nielsen, Kolmogorov's Mapping Neural Network Existence Theorem, First IEEE International Conference on Neural Networks, San Diego, Vol. 3, 1987
> > > [3] V. Kůrková, Kolmogorov's Theorem Is Relevant, Neural computation 3 (4), 617-622, 1991
> > >
> > > **Negative results.** Your arguments seem to primarily address Theorem 5. But we also have "negative" theorems 2, 3, 4, 7, 9,  that actually *confirm* your point of pathology by clarifying precisely in which sense this pathology is manifested. These are new, rigorous and not so obvious results that constitute a significant part of our contribution.
> > >
> > > > I do not know of any examples in the statistics/ML literature where one is interested in *exactly* fitting a dimension $d$ model with $W<d$ parameters.
> > >
> > > If we are interested only in a non-exact fitting, say with accuracy $\delta>0$, then a respective GF-learnable 2-parameter model can be easily constructed by truncating the model $\Phi$ presented in Theorem 5. Such a truncated model is no longer pathological, in the sense that it is now expressible by an elementary function, say in the form of some finite neural network (Theorem 7 no longer applies). We don't claim that such a model is *practically useful*, but our construction shows that it is theoretically possible while being relatively realistic.

---

### Official Review · Reviewer_hbxS · 2024-07-12

**Soundness:** 4
**Presentation:** 3
**Contribution:** 3
**Rating:** 8
**Confidence:** 3

**Summary:**

The paper studies the *learnability* of finite dimensional space of functions $\mathcal{H}$ of dimension $d$. The learnability criterion is related to the gradient flow according to the canonical $L^2$ error, associated with a certain $\Phi$-parametrized family of dimension $W$, that can be chosen. The authors proposes the following results:

- **Theorem 3**: For $W < d$, under some regularity assumption on $\Phi$, there exists a ball in $\mathcal{H}$ with non-reachable function. It implies that under this condition on the choice of $\Phi$-space, the GF-learnable function space is not dense in $\mathcal{H}$.

- **Theorem 4**: Any subset homeomorphic to the $W$-sphere contains non learnable targets.

- **Theorem 5**: On the positive side, there exists for $W = 2$ models that can learn targets with probability arbitrarily close to $0$.

- **Results 7-9**: Finally the authors try to answer the question of learnability of functions expressible with elementary operations ("simple functions"). For this, Pfaffian maps are defined and for underparametrized Pfaffian models, GF-reachable functions have Lebesgue-measure $0$.

**Strengths:**

The paper has the following strengths:

- The presentation is very clear, even-though the theme of the study is difficult and technical. The proofs presented are clear and sharp, the main Theorem is illustrated and the idea of the construction is nicely displayed. Finally, the overall questioning of the authors is clear and very nicely presented into a self-contained story. This is a very nice article to read!

- All the results try to give a picture of the set of function that is GF-learnable: it is a (super) hard task and yet, the authors tackle it very elegantly.

- Technically the results *seem* very strong.

**Weaknesses:**

Reviewing such a paper in a conference (along with 5 others) is a very difficult task given the time constraint and the technicality of the present paper. Hence, I apologize in advance if the questions I am raising are due to the limit of my knowledge and not the lack of clarity of the authors. Yet, considering that I might be a typical reader of the article I want to mention the following points that could help improving the manuscript:

The statements given by theorems 3 and 4 on the one side and theorem 5 on the other side seem to go in two opposite directions, at least qualitatively: indeed Theorems 3 and 4 go in the direction that there always exists non-reachable functions (even an open Ball for Thm 3!), whereas Theorem 5 argues that if we equip the space of function with any Borel measure, GF-learnability can be almost surely certified. Althought, the two statement do not contradict themselves, it would be good to comment more on this fact: under this form, it does not help me picturing exactly what the set of learnable function look like. For example, if we restrict the space of functions to a compact subset of $\mathcal{H}$, doesn't Theorem 5 implies a sort of density of learnable functions?

**Questions:**

See above for an important first question.

-  line 298, and line 195, the authors argue that the consequence of Theorem 4 is that the set of non-learnable function is dense for $W < d$. As far as I felt it, it seems to be a direct consequence of the fact that a neighborhood of some $f$ always contains a set homeomorphic to the W-dimensional sphere, but
    - it would be better to have a clear (even if two-line) proof
    - that would help the reader to have a clear **corollary** to stamp this property

- I would change the title to * **Learnability** of high-dimensional targets by two-parameter models with gradient flow*  to put emphasis that the authors do not study a specific class of models but more whether certain functions are not learnable.

Minor additionnal comments:

- l. 68-69: I do not see why Theorem 1 is *a low-dimensional reduction* reflected in Theorem 1.
- l. 172: $J^*_0 J_0$ instead of $J^*_0J$
- l.252 : in which

---

> ### Author Rebuttal · Authors · 2024-08-07
>
> Thank you for the useful feedback and a very positive evaluation of our work!
>
> * "*What the set of learnable functions look like... Doesn't Theorem 5 implies a sort of density of learnable functions?*"
>
> The right geometric picture for the set of learnable functions in Theorem 5 would be a multidimensional "fat Cantor set" (a.k.a. Smith–Volterra–Cantor set), see e.g. [this Wikipedia article](https://en.wikipedia.org/wiki/Smith%E2%80%93Volterra%E2%80%93Cantor_set) and the illustrations there. Such sets are not dense in the ambient space, but can be quite large in terms of their measure. This is exactly what happens in Theorem 5.
>
> It is an interesting question whether the set of learnable targets may need to actually have more "holes" than predicted by Theorems 3 and 4. In particular, Theorem 3 only constructs a single ball devoid of learnable targets. However, the learnable set constructed in Theorem 5 is even nowhere dense, i.e. an arbitrary ball in the target space contains a ball devoid of learnable targets - a strictly stronger property. In this sense, our results are not tight.
>
> * "*It would help the reader to have a clear corollary..  that the set of non-learnable function is dense for $W<d$*"
>
> Indeed, such a corollary would be quite reasonable, thank you for this suggestion.
>
> * "*I would change the title to **Learnability of ...** *"
>
> Yes, this is also quite reasonable, thank you.
>
> * "*I do not see why Theorem 1 is a low-dimensional reduction reflected in Theorem 1*"
>
> This sentence is indeed somewhat awkward and probably needs to be clarified. We just meant to say that Theorem 1 gives an example of a reduction of a high-dimensional target space to a low-dimensional parametric description, and we ask if such or similar reduction can be combined with learning by GD.

---

> > ### Comment · Reviewer_hbxS · 2024-08-10
> > **After Rebuttal**
> >
> > I thank the authors for the rebuttal. I still find the article a very elegant contribution and keep my score.

---

### Official Review · Reviewer_XxKo · 2024-07-13

**Soundness:** 3
**Presentation:** 2
**Contribution:** 2
**Rating:** 6
**Confidence:** 2

**Summary:**

This paper analyzes when it is possible to define a map $\Phi: \mathbb{R}^W \to \mathbb{R}^d$ such that any point $y \in \mathbb{R}^d$ is ``learnable'' via gradient flow on the square loss $\|y - \Phi(w)\|^2$. They show that for any distribution over $y$, there exists a map $\Phi: \mathbb{R}^2 \to \mathbb{R}^d$ such that gradient flow on the square loss gets arbitrarily close to any $y$ with probability arbitrarily close to $1$. It also shows that it is impossible for this to hold with probability exactly $1$ for any $W < d$, and shows that in this case the non-learnable targets are dense in $\mathbb{R}^d$.

**Strengths:**

- The existence of the map $\Phi$ in Theorem 5 is surprising, especially given the less surprising results in Theorems 3,4.
- The impossibility results in Theorems 3,4 are clear and well presented.

**Weaknesses:**

- The construction and proof sketch for Theorem 5 and the diagrams in Figure 2 are very difficult to follow.
- It is strange to define the setting in terms of abstract Hilbert spaces only to immediately specialize to the case of $\mathbb{R}^d$. In the supervised learning analogy, this corresponds to the uniform measure over a finite sample of $n = d$ datapoints.

**Questions:**

- Why is it necessary to restrict to $\mathcal{H} = \mathbb{R}^d$? Is it possible to extend the arguments to infinite dimensional spaces?
- Is it possible to simplify the construction in Theorem 5 for some easy target distributions to simplify exposition?

**Limitations:**

The authors have adequately addressed the limitations of their work.

---

> ### Author Rebuttal · Authors · 2024-08-07
>
> Thank you for your careful reading and a positive evaluation of our work!
>
> * "*The construction and proof sketch for Theorem 5 and the diagrams in Figure 2 are very difficult to follow.*"
>
> Thank you for this feedback. We admit that some elements of the proof sketch may be not clear enough. We wanted to convey the key ideas, but due to the size constraints we found it hard to give more than a flavor of the proof. We'll try to improve this sketch. Meanwhile, please see Appendix A, where the full proof is carefully described.
>
> * "*Is it possible to extend the arguments to infinite dimensional spaces?*"
>
> This is an interesting question. In fact, an inspection of our "negative" Theorems 2-4 shows that they remain valid if the target space is an infinite-dimensional Hilbert space while the number of parameters is finite. This is natural because, clearly, the addressed problem becomes harder if the dimensionality of the target space increases.
>
> Following this logic, the negative Theorem 7 about elementary functions can probably also be generalized to infinite dimensions, though in this case some reformulation is definitely required because there is no Lebesgue measure on the infinite-dimensional Hilbert space.
>
> Our main "positive" Theorem 5 can probably also be extended to infinite dimensions, but such an extension has some aspects that we haven't figured out; please see our [general response](https://openreview.net/forum?id=8XoWofmZkI&noteId=qNVXBofICV).
>
> * "*Is it possible to simplify the construction in Theorem 5 for some easy target distributions?*"
>
> The natural simple target distribution is just the standard uniform distribution in a box. Our construction for a general distribution is only a small modification of the construction for this special case. However, this means that all the essential elements of the construction are present in this special case and so, unfortunately, it is not much easier than the general case.

---

> > ### Comment · Reviewer_XxKo · 2024-08-10
> >
> > Thank you for addressing my questions and concerns. I still believe that while the paper is well written and the results are surprising, the paper is held back by a lack of intuition for the main result (Theorem 5). I have decided to keep my score.

---

### Official Review · Reviewer_MmYK · 2024-07-13

**Soundness:** 2
**Presentation:** 1
**Contribution:** 2
**Rating:** 5
**Confidence:** 2

**Summary:**

This submission studies the learnability of high-dimensional targets with models having fewer parameters. The manuscript considers training with Gradient Flow and studies when the target -- identified by a general d-dimensional probability distribution -- can be learned by $W-$ parameter models, with $W <d$. First, the authors provably show general impossibility results, i.e., there is always a substantial amount of non-learnable targets (with the extreme case to be zero-Lebesgue measure for $W=1$). However, they prove also positive results regarding GF learnability with $W=2$: it is always possible to find a map $\Phi$ for which the learnable set can be made arbitrarily large.

**Strengths:**

The main strength of the present submission is the strong theoretical apparatus built to prove the desired results. From the text, the authors specify what results are conjectured and what are rigorously proven. On the other hand, the formal presentation sometimes penalizes the readability of the text (see the section below).

**Weaknesses:**

The weakness of this paper is the clarity of the exposition. I believe that many parts of the main body could be moved to the appendix to make the submission more readable for non-expert readers. This comment is not set to undermine the quality of the contribution, which has nice theoretical aspects, but to enhance the presentation of the main results. See the section below for pointers.

**Questions:**

- As described above I believe that many formalities should be moved to the appendix to be accessible for the NeurIPS audience. My main concern is on Section 5, which I believe to be interesting, but hard to read. I would enlarge the discussion part of this section considering that only two lines are used after corollary 9.
- How would the present construction relate to general kernel /random feature methods? The mathematical apparatus is similar (RKHS), but with different purposes. I believe connection with classical machine-learning tools would help the typical NeurIPS reader.  (Page 4 for example).
- Maybe the authors could use also polynomial targets in their "Examples" paragraph (which are more than welcome to help the reader)? More generally, non-linear targets would help to see the extent of the present setting.
- Am I correct to understand that general non-linear function are not admissible in the present setting? If that is the case, stating it explicitly would help the reader.
- I feel like the nomenclature for target-dimension ad $d$ is confusing as one might be led to think that is the output dimension (Y). Maybe it could be emphasized more in the text.
- Below Eq. (1) I would expand on what is meant to be "locally solvable" given that the whole paper will turn around the GF setting.
- Is there a connection of your results to the recent work [1]? I acknowledge that the reference was only recently put on arxiv and I am just wondering out of curiosity for this minor point. In any case, I think it would be good to mention that ideas using KST are recently used in the machine-learning community.
- How important is in your work the condition $w(0) = 0$? What would it change if the weights could be somehow correlated with the target's ones?


[1] KAN: Kolmogorov–Arnold Networks. Liu et al. 2024. arXiv:2404.19756.

**Limitations:**

The limitations are discussed in the submission.

---

> ### Author Rebuttal · Authors · 2024-08-07
>
> Thank you for the careful reading of our paper and many useful comments.
>
> * "*I would enlarge the discussion part [of Section 5]*"
>
> Thank you for this feedback. Indeed, we agree that it can be useful to add some comments and maybe a graphic illustration here.
>
> * "*Am I correct to understand that general non-linear function are not admissible in the present setting?*"
>
> To clarify, we assume the target space $\mathcal H$ to be a linear space of functions, but the functions themselves need not be linear (or have any other particular form). Examples include those we mention in lines 111-119 (the full $L^2$ space, linear targets, polynomial targets) and RKHS subspaces associated with finite sets of data points (see below). Our results only use the euclidean structure of the space $\mathcal H$ (i.e., as a linear space with a scalar product). Representation of targets in $\mathcal H$ as particular functions is the starting point of our study, but it only affects the GF flow through this euclidean structure. In other words, once we know the scalar products between targets, their other properties do not play any role in our theorems.
>
> * "*Maybe the authors could use also polynomial targets in their "Examples" paragraph*"
>
> Sure, we do that (our third example).
>
> * "*How would the present construction relate to general kernel /random feature methods?*"
>
> One can consider a subspace of RKHS as our target space $\mathcal H$. Specifically, suppose we have a kernel $K(\cdot, \cdot)$ on the input space $X$, and a finite training set $(\mathbf x_n, y_n)_{n=1}^N$ representing some function $f:X\to \mathbb R$.
>
> In kernel methods, the function $f$ is fitted by functions $\widetilde f_{\mathbf c}(\mathbf x)=\sum_{k=1}^N c_k K(\mathbf x,\mathbf x_k),$ with some coefficients $\mathbf c=(c_1,\ldots,c_N)$. We can naturally view such functions $\widetilde f_{\mathbf c}$, with various $\mathbf c$, as forming our target space $\mathcal H$. Assuming the kernel is non-degenerate, $\mathcal H$ is $N$-dimensional, with vectors uniquely corresponding to the coefficient vectors $\mathbf c$. Then Theorem 5 shows that, by using gradient flow with a suitable two-parameter map $\Phi:\mathbb R^2\to\mathcal H$, we can, with a high probability, learn the target $\widetilde f_{\mathbf c^*}$ that interpolates the training data $(\mathbf x_n, y_n)_{n=1}^N$.
>
> * "*I feel like the nomenclature for target-dimension as $d$ is confusing*"
>
> Thanks for this feedback; we'll clarify this point.
>
> * "*Below Eq. (1) I would expand on what is meant to be "locally solvable"*"
>
> Thanks for this feedback, we'll clarify that ("locally solvable" means solvable at least for a finite interval of times).
>
> * "*Is there a connection of your results to the recent work [1]?*"
>
> Yes, we are familiar with that work. Both that work and our paper have a common general theme of a low-dimensional reduction (that we alluded to in our introduction), but otherwise the connection is fairly weak. The work in question proposes a KST-inspired architecture and focuses on its expressivity, whereas our focus is on GF optimization. Also, our construction is rather different from the one used in KST.
>
> * "*How important is in your work the condition* $\mathbf w(0)=0$?  *What would it change if the weights could be somehow correlated with the target's ones?*"
>
> The condition that GF starts from $\mathbf w(0)=0$ is not important, one could instead start GF from any fixed $\mathbf w_0$.
>
> Regarding a target-dependent initial condition, it's an interesting question. Some of our "negative" results such as Theorems 2 and 3 do not apply in this setting (at least without some essential modifications). On the other hand, the negative Theorem 4  based on Borsuk-Ulam remains valid as long as the initial condition $\mathbf w_0$ depends continuously on the target. An important difference would be in our main positive Theorem 5: as we mention in the discussion, we have not been able to prove it for infinite measures such as the Lebegue measure. However, this can certainly be done with a target-dependent initial condition: we can just split the space into finite boxes, assign a separate initial condition for each box, and define the map $\Phi$ separately in each respective region of the parameter space by following the existing proof for a finite box.

---

> > ### Comment · Reviewer_MmYK · 2024-08-10
> > **Thank you for your rebuttal**
> >
> > I thank the authors for their rebuttal that clarified my concerns. After carefully reading the authors' responses along with other reviewers’ comments, I believe the proposed changes will improve the quality of the submission. I would like to keep my score as in the original review.

---

### Author Rebuttal · Authors · 2024-08-07

We sincerely thank the reviewers for a careful reading of our paper and many useful comments and suggestions.

Reviewers XxKo and qWpD ask about a possible extension of our main Theorem 5 to infinite-dimensional target spaces. We believe that this can be done, but there are some subtleties that we haven't worked out yet.

One immediate obstacle is as follows. Our finite-dimensional proof very much relies on the fact that we can choose in the target space a box $[-c,c]^d$ having measure $1-\epsilon$. The Cartesian product structure of the box is crucial for the proof. In the infinite-dimensional setting, the natural analog would be a box $\times_{k=1}^\infty [-c_k, c_k]$ in the Hilbert space $l^2$. For this box to lie in $l^2$, we need the convergence $\sum_{k=1}^\infty c_k^2<\infty$. However, we haven't managed to prove that such a box of measure $1-\epsilon$ can be found for an arbitrary Borel (with respect to the norm topology) measure on $l^2$. So either this has to be additionally proved, or the theorem must be stated for a more restrictive set of measures than arbitrary Borel measures considered in Theorem 5.

---

### Comment · Area_Chair_4dJs · 2024-08-08

Just a friendly reminder to the reviewers to acknowledge the author's rebuttal, so that the discussion period can be used efficiently.

---

### Decision · Program_Chairs · 2024-09-25

**Decision:**

Accept (poster)

**Comment:**

The paper investigates an interesting and novel question with mathematical rigor. Most reviews all lean towards acceptance.